# Large-scale animal model study uncovers altered brain pH and lactate levels as a transdiagnostic endophenotype of neuropsychiatric disorders involving cognitive impairment

Hideo Hagihara[1]*, Hirotaka Shoji[1], Satoko Hattori[1], Giovanni Sala[1], Yoshihiro Takamiya[1], Mika Tanaka[1], Masafumi Ihara[2], Mihiro Shibutani[3], Izuho Hatada[3], Kei Hori[4], Mikio Hoshino[4], Akito Nakao[5], Yasuo Mori[5], Shigeo Okabe[6], Masayuki Matsushita[7], Anja Urbach[8], Yuta Katayama[9], Akinobu Matsumoto[9], Keiichi I Nakayama[9], Shota Katori[10], Takuya Sato[10], Takuji Iwasato[10], Haruko Nakamura[11], Yoshio Goshima[11], Matthieu Raveau[12], Tetsuya Tatsukawa[12], Kazuhiro Yamakawa[12,13], Noriko Takahashi[14,15], Haruo Kasai[14,16], Johji Inazawa[17], Ikuo Nobuhisa[18], Tetsushi Kagawa[18], Tetsuya Taga[18], Mohamed Darwish[19,20], Hirofumi Nishizono[21], Keizo Takao[20,22], Kiran Sapkota[23], Kazutoshi Nakazawa[23], Tsuyoshi Takagi[24], Haruki Fujisawa[25], Yoshihisa Sugimura[25], Kyosuke Yamanishi[26], Lakshmi Rajagopal[27], Nanette Deneen Hannah[27], Herbert Y Meltzer[27], Tohru Yamamoto[28], Shuji Wakatsuki[29], Toshiyuki Araki[29], Katsuhiko Tabuchi[30], Tadahiro Numakawa[31], Hiroshi Kunugi[31,32], Freesia L Huang[33], Atsuko Hayata-Takano[34,35,36], Hitoshi Hashimoto[34,36,37,38,39], Kota Tamada[40,41], Toru Takumi[40,41], Takaoki Kasahara[42,43], Tadafumi Kato[42,44], Isabella A Graef[45], Gerald R Crabtree[45], Nozomi Asaoka[46], Hikari Hatakama[47], Shuji Kaneko[47], Takao Kohno[48], Mitsuharu Hattori[48], Yoshio Hoshiba[49], Ryuhei Miyake[50], Kisho Obi-Nagata[50], Akiko Hayashi-Takagi[49,50], Léa J Becker[51], Ipek Yalcin[51], Yoko Hagino[52], Hiroko Kotajima-Murakami[52], Yuki Moriya[52], Kazutaka Ikeda[52], Hyopil Kim[53,54], Bong-Kiun Kaang[53,55], Hikari Otabi[56,57], Yuta Yoshida[56], Atsushi Toyoda[56,57,58], Noboru H Komiyama[59,60], Seth GN Grant[59,60], Michiru Ida-Eto[61], Masaaki Narita[61], Ken-ichi Matsumoto[62], Emiko Okuda-Ashitaka[63], Iori Ohmori[64], Tadayuki Shimada[65], Kanato Yamagata[65], Hiroshi Ageta[66], Kunihiro Tsuchida[66], Kaoru Inokuchi[67,68,69], Takayuki Sassa[70], Akio Kihara[70], Motoaki Fukasawa[71], Nobuteru Usuda[71], Tayo Katano[72], Teruyuki Tanaka[73], Yoshihiro Yoshihara[74], Michihiro Igarashi[75,76], Takashi Hayashi[77], Kaori Ishikawa[78,79], Satoshi Yamamoto[80], Naoya Nishimura[80], Kazuto Nakada[78,79], Shinji Hirotsune[81], Kiyoshi Egawa[82], Kazuma Higashisaka[83], Yasuo Tsutsumi[83], Shoko Nishihara[84], Noriyuki Sugo[85], Takeshi Yagi[85], Naoto Ueno[86], Tomomi Yamamoto[87], Yoshihiro Kubo[87], Rie Ohashi[88,89,90], Nobuyuki Shiina[88,89,90], Kimiko Shimizu[91], Sayaka Higo-Yamamoto[92], Katsutaka Oishi[92,93,94,95], Hisashi Mori[96], Tamio Furuse[97], Masaru Tamura[97], Hisashi Shirakawa[98], Daiki X Sato[1,99], Yukiko U Inoue[4], Takayoshi Inoue[4], Yuriko Komine[100,101], Tetsuo Yamamori[101,102], Kenji Sakimura[103,104], Tsuyoshi Miyakawa[1]*

[1]Division of Systems Medical Science, Center for Medical Science, Fujita Health University, Toyoake, Japan; [2]Department of Neurology, National Cerebral and

*For correspondence:
h-hagi@fujita-hu.ac.jp (HH);
miyakawa@fujita-hu.ac.jp (TM)

Cardiovascular Center, Suita, Japan; [3]Laboratory of Genome Science, Biosignal Genome Resource Center, Institute for Molecular and Cellular Regulation, Gunma University, Maebashi, Japan; [4]Department of Biochemistry and Cellular Biology, National Institute of Neuroscience, National Center of Neurology and Psychiatry, Kodaira, Japan; [5]Department of Synthetic Chemistry and Biological Chemistry, Graduate School of Engineering, Kyoto University, Kyoto, Japan; [6]Department of Cellular Neurobiology, Graduate School of Medicine, The University of Tokyo, Tokyo, Japan; [7]Department of Molecular Cellular Physiology, Graduate School of Medicine, University of the Ryukyus, Nishihara, Japan; [8]Department of Neurology, Jena University Hospital, Jena, Germany; [9]Department of Molecular and Cellular Biology, Medical Institute of Bioregulation, Kyushu University, Fukuoka, Japan; [10]Laboratory of Mammalian Neural Circuits, National Institute of Genetics, Mishima, Japan; [11]Department of Molecular Pharmacology and Neurobiology, Yokohama City University Graduate School of Medicine, Yokohama, Japan; [12]Laboratory for Neurogenetics, RIKEN Center for Brain Science, Wako, Japan; [13]Department of Neurodevelopmental Disorder Genetics, Institute of Brain Sciences, Nagoya City University Graduate School of Medical Sciences, Nagoya, Japan; [14]Laboratory of Structural Physiology, Center for Disease Biology and Integrative Medicine, Faculty of Medicine, The University of Tokyo, Tokyo, Japan; [15]Department of Physiology, Kitasato University School of Medicine, Sagamihara, Japan; [16]International Research Center for Neurointelligence (WPI-IRCN), UTIAS, The University of Tokyo, Tokyo, Japan; [17]Research Core, Tokyo Medical and Dental University, Tokyo, Japan; [18]Department of Stem Cell Regulation, Medical Research Institute, Tokyo Medical and Dental University, Tokyo, Japan; [19]Department of Biochemistry, Faculty of Pharmacy, Cairo University, Cairo, Egypt; [20]Department of Behavioral Physiology, Graduate School of Innovative Life Science, University of Toyama, Toyama, Japan; [21]Medical Research Institute, Kanazawa Medical University, Kahoku, Japan; [22]Department of Behavioral Physiology, Faculty of Medicine, University of Toyama, Toyama, Japan; [23]Department of Neuroscience, Southern Research, Birmingham, United States; [24]Institute for Developmental Research, Aichi Developmental Disability Center, Kasugai, Japan; [25]Department of Endocrinology, Diabetes and Metabolism, School of Medicine, Fujita Health University, Toyoake, Japan; [26]Department of Neuropsychiatry, Hyogo Medical University School of Medicine, Nishinomiya, Japan; [27]Department of Psychiatry and Behavioral Sciences, Northwestern University Feinberg School of Medicine, Chicago, United States; [28]Department of Molecular Neurobiology, Faculty of Medicine, Kagawa University, Kita-gun, Japan; [29]Department of Peripheral Nervous System Research, National Institute of Neuroscience, National Center of Neurology and Psychiatry, Tokyo, Japan; [30]Department of Molecular & Cellular Physiology, Shinshu University School of Medicine, Matsumoto, Japan; [31]Department of Mental Disorder Research, National Institute of Neuroscience, National Center of Neurology and Psychiatry, Kodaira, Japan; [32]Department of Psychiatry, Teikyo University School of Medicine, Tokyo, Japan; [33]Program of Developmental Neurobiology, National Institute of Child Health and Human Development, National Institutes of Health, Bethesda, United States; [34]Laboratory of Molecular Neuropharmacology, Graduate School of Pharmaceutical Sciences, Osaka University, Suita, Japan; [35]Department of Pharmacology, Graduate School of Dentistry, Osaka University, Suita, Japan; [36]United Graduate School of Child Development, Osaka University, Kanazawa University, Hamamatsu University School of Medicine, Chiba University and University of Fukui, Suita, Japan; [37]Division of Bioscience, Institute for Datability Science, Osaka University, Suita, Japan; [38]Transdimensional Life Imaging Division, Institute for Open and Transdisciplinary Research Initiatives, Osaka University, Suita, Japan;

[39]Department of Molecular Pharmaceutical Science, Graduate School of Medicine, Osaka University, Suita, Japan; [40]RIKEN Brain Science Institute, Wako, Japan; [41]Department of Physiology and Cell Biology, Kobe University School of Medicine, Kobe, Japan; [42]Laboratory for Molecular Dynamics of Mental Disorders, RIKEN Center for Brain Science, Wako, Japan; [43]Institute of Biology and Environmental Sciences, Carl von Ossietzky University of Oldenburg, Oldenburg, Germany; [44]Department of Psychiatry and Behavioral Science, Juntendo University Graduate School of Medicine, Tokyo, Japan; [45]Department of Pathology, Stanford University School of Medicine, Stanford, United States; [46]Department of Pharmacology, Kyoto Prefectural University of Medicine, Kyoto, Japan; [47]Department of Molecular Pharmacology, Graduate School of Pharmaceutical Sciences, Kyoto University, Kyoto, Japan; [48]Department of Biomedical Science, Graduate School of Pharmaceutical Sciences, Nagoya City University, Nagoya, Japan; [49]Laboratory of Medical Neuroscience, Institute for Molecular and Cellular Regulation, Gunma University, Maebashi, Japan; [50]Laboratory for Multi-scale Biological Psychiatry, RIKEN Center for Brain Science, Wako, Japan; [51]Institut des Neurosciences Cellulaires et Intégratives, Centre National de la Recherche Scientifique, Université de Strasbourg, Strasbourg, France; [52]Addictive Substance Project, Tokyo Metropolitan Institute of Medical Science, Tokyo, Japan; [53]Department of Biological Sciences, College of Natural Sciences, Seoul National University, Seoul, Republic of Korea; [54]Department of Biomedical Engineering, Johns Hopkins School of Medicine, Baltimore, United States; [55]Center for Cognition and Sociality, Institute for Basic Science (IBS), Daejeon, Republic of Korea; [56]College of Agriculture, Ibaraki University, Ami, Japan; [57]United Graduate School of Agricultural Science, Tokyo University of Agriculture and Technology, Fuchu, Japan; [58]Ibaraki University Cooperation between Agriculture and Medical Science (IUCAM), Ibaraki, Japan; [59]Genes to Cognition Program, Centre for Clinical Brain Sciences, University of Edinburgh, Edinburgh, United Kingdom; [60]Simons Initiative for the Developing Brain, Centre for Discovery Brain Sciences, University of Edinburgh, Edinburgh, United Kingdom; [61]Department of Developmental and Regenerative Medicine, Mie University, Graduate School of Medicine, Tsu, Japan; [62]Department of Biosignaling and Radioisotope Experiment, Interdisciplinary Center for Science Research, Organization for Research and Academic Information, Shimane University, Izumo, Japan; [63]Department of Biomedical Engineering, Osaka Institute of Technology, Osaka, Japan; [64]Department of Physiology, Okayama University Graduate School of Medicine, Dentistry and Pharmaceutical Sciences, Okayama, Japan; [65]Child Brain Project, Tokyo Metropolitan Institute of Medical Science, Tokyo, Japan; [66]Division for Therapies Against Intractable Diseases, Center for Medical Science, Fujita Health University, Toyoake, Japan; [67]Research Center for Idling Brain Science, University of Toyama, Toyama, Japan; [68]Department of Biochemistry, Graduate School of Medicine and Pharmaceutical Sciences, University of Toyama, Toyama, Japan; [69]Core Research for Evolutionary Science and Technology (CREST), Japan Science and Technology Agency (JST), University of Toyama, Toyama, Japan; [70]Faculty of Pharmaceutical Sciences, Hokkaido University, Sapporo, Japan; [71]Department of Anatomy II, Fujita Health University School of Medicine, Toyoake, Japan; [72]Department of Medical Chemistry, Kansai Medical University, Hirakata, Japan; [73]Department of Developmental Medical Sciences, Graduate School of Medicine, The University of Tokyo, Tokyo, Japan; [74]Laboratory for Systems Molecular Ethology, RIKEN Center for Brain Science, Wako, Japan; [75]Department of Neurochemistry and Molecular Cell Biology, School of Medicine, and Graduate School of Medical and Dental Sciences, Niigata University, Niigata, Japan; [76]Transdiciplinary Research Program, Niigata University, Niigata,

Japan; [77]Biomedical Research Institute, National Institute of Advanced Industrial Science and Technology (AIST), Tsukuba, Japan; [78]Institute of Life and Environmental Sciences, University of Tsukuba, Tsukuba, Japan; [79]Graduate School of Science and Technology, University of Tsukuba, Tsukuba, Japan; [80]Integrated Technology Research Laboratories, Pharmaceutical Research Division, Takeda Pharmaceutical Company, Ltd, Fujisawa, Japan; [81]Department of Genetic Disease Research, Osaka City University Graduate School of Medicine, Osaka, Japan; [82]Department of Pediatrics, Hokkaido University Graduate School of Medicine, Sapporo, Japan; [83]Laboratory of Toxicology and Safety Science, Graduate School of Pharmaceutical Sciences, Osaka University, Suita, Japan; [84]Glycan & Life Systems Integration Center (GaLSIC), Soka University, Tokyo, Japan; [85]Graduate School of Frontier Biosciences, Osaka University, Suita, Japan; [86]Laboratory of Morphogenesis, National Institute for Basic Biology, Okazaki, Japan; [87]Division of Biophysics and Neurobiology, National Institute for Physiological Sciences, Okazaki, Japan; [88]Laboratory of Neuronal Cell Biology, National Institute for Basic Biology, Okazaki, Japan; [89]Department of Basic Biology, SOKENDAI (Graduate University for Advanced Studies), Okazaki, Japan; [90]Exploratory Research Center on Life and Living Systems (ExCELLS), National Institutes of Natural Sciences, Okazaki, Japan; [91]Department of Biological Sciences, School of Science, The University of Tokyo, Tokyo, Japan; [92]Healthy Food Science Research Group, Cellular and Molecular Biotechnology Research Institute, National Institute of Advanced Industrial Science and Technology (AIST), Tsukuba, Japan; [93]Department of Applied Biological Science, Graduate School of Science and Technology, Tokyo University of Science, Noda, Japan; [94]Department of Computational Biology and Medical Sciences, Graduate School of Frontier Sciences, The University of Tokyo, Kashiwa, Japan; [95]School of Integrative and Global Majors (SIGMA), University of Tsukuba, Tsukuba, Japan; [96]Department of Molecular Neuroscience, Graduate School of Medicine and Pharmaceutical Sciences, University of Toyama, Toyama, Japan; [97]Mouse Phenotype Analysis Division, Japan Mouse Clinic, RIKEN BioResource Research Center (BRC), Tsukuba, Japan; [98]Department of Molecular Pharmacology, Graduate School of Pharmaceutical Sciences, Kyoto University, Kyoto, Japan; [99]Graduate School of Life Sciences, Tohoku University, Sendai, Japan; [100]Young Researcher Support Group, Research Enhancement Strategy Office, National Institute for Basic Biology, National Institute of Natural Sciences, Okazaki, Japan; [101]Division of Brain Biology, National Institute for Basic Biology, Okazaki, Japan; [102]Laboratory for Molecular Analysis of Higher Brain Function, RIKEN Center for Brain Science, Wako, Japan; [103]Department of Cellular Neurobiology, Brain Research Institute, Niigata University, Niigata, Japan; [104]Department of Animal Model Development, Brain Research Institute, Niigata University, Niigata, Japan

**Abstract** Increased levels of lactate, an end-product of glycolysis, have been proposed as a potential surrogate marker for metabolic changes during neuronal excitation. These changes in lactate levels can result in decreased brain pH, which has been implicated in patients with various neuropsychiatric disorders. We previously demonstrated that such alterations are commonly observed in five mouse models of schizophrenia, bipolar disorder, and autism, suggesting a shared endophenotype among these disorders rather than mere artifacts due to medications or agonal state. However, there is still limited research on this phenomenon in animal models, leaving its generality across other disease animal models uncertain. Moreover, the association between changes in brain lactate levels and specific behavioral abnormalities remains unclear. To address these gaps, the International Brain pH Project Consortium investigated brain pH and lactate levels in 109 strains/conditions of 2294 animals with genetic and other experimental manipulations relevant

to neuropsychiatric disorders. Systematic analysis revealed that decreased brain pH and increased lactate levels were common features observed in multiple models of depression, epilepsy, Alzheimer's disease, and some additional schizophrenia models. While certain autism models also exhibited decreased pH and increased lactate levels, others showed the opposite pattern, potentially reflecting subpopulations within the autism spectrum. Furthermore, utilizing large-scale behavioral test battery, a multivariate cross-validated prediction analysis demonstrated that poor working memory performance was predominantly associated with increased brain lactate levels. Importantly, this association was confirmed in an independent cohort of animal models. Collectively, these findings suggest that altered brain pH and lactate levels, which could be attributed to dysregulated excitation/inhibition balance, may serve as transdiagnostic endophenotypes of debilitating neuropsychiatric disorders characterized by cognitive impairment, irrespective of their beneficial or detrimental nature.

## eLife assessment

The manuscript offers **useful** descriptive insights into the potential influence of whole-brain lactate and pH levels on the manifestation of behavioral phenotypes seen in diverse animal models of neuropsychiatric disorders. However, reviewers have raised concerns about the potential loss of specificity in capturing regional and cell-type-specific effects when relying solely on whole-brain analysis methods. While the evidence supporting the conclusions is largely **solid**, the robustness of these conclusions could be enhanced by the inclusion of additional data and further analysis.

## Introduction

Neuropsychiatric disorders, such as schizophrenia (SZ), bipolar disorder (BD), major depressive disorder (MDD), autism spectrum disorder (ASD), and Alzheimer's disease (AD), are relatively common. More than one-third of the population in most countries is diagnosed with at least one of these disorders at some point in their life (*WHO International Consortium in Psychiatric Epidemiology, 2000*). Although these diseases are characterized by different clinical diagnostic categories, they share some biological features, such as genetic mutations, molecular changes, and brain activity alterations (*Argyelan et al., 2014*; *Cardno and Owen, 2014*; *Douaud et al., 2014*; *Forero et al., 2016*; *Hall et al., 2015*), suggesting a common underlying biological basis. Accumulating evidence suggests that metabolic changes in the brain are common to several neuropsychiatric disorders. Increased levels of lactate, an end product of the glycolysis pathway, have been observed in the brains of patients with SZ, BD, ASD, MDD, and epilepsy (*Dager et al., 2004*; *Goh et al., 2014*; *Greene et al., 2003*; *Halim et al., 2008*; *Machado-Vieira et al., 2017*; *Prabakaran et al., 2004*; *Rossignol and Frye, 2012*; *Rowland et al., 2016*; *Soeiro-de-Souza et al., 2016*; *Sullivan et al., 2019*). Brain lactate levels have been observed to rise during neuronal excitation induced by somatic stimuli (*Koush et al., 2019*; *Mangia et al., 2007*) and epileptic seizures (*During et al., 1994*; *Lazeyras et al., 2000*; *Najm et al., 1997*; *Siesjö et al., 1985b*). This increase is accompanied by a concurrent elevation in the excitatory neurotransmitter glutamate (*Fernandes et al., 2020*; *Schaller et al., 2014*; *Schaller et al., 2013*). Additionally, lactate has been found to increase the excitability of several populations of neurons (*Magistretti and Allaman, 2018*). The presence of increased brain lactate levels aligns with the neuronal hyperexcitation hypothesis proposed for neuropsychiatric disorders, such as SZ (*Heckers and Konradi, 2015*; *Whitfield-Gabrieli et al., 2009*), BD (*Chen et al., 2011*; *Mertens et al., 2015*), and AD (*Bi et al., 2020*; *Busche and Konnerth, 2015*).

Increased lactate levels decrease tissue pH, which may also be associated with deficits in brain energy metabolism (*Prabakaran et al., 2004*). Lactate is a relatively strong acid and is almost completely dissociated into $H^+$ ions and lactate anions at cellular pH (*Siesjö, 1985a*). Furthermore, $H^+$ ions are one of the most potent intrinsic neuromodulators in the brain in terms of concentration and thus play an important role in the control of gene expression (*Hagihara et al., 2023*; *Mexal et al., 2006*; *Mistry and Pavlidis, 2010*) and cellular functions of neurons and glial cells (*Chesler, 2003*; *Kaila and Ransom, 1998*). Recent meta-analyses have confirmed decreased brain pH and increased lactate levels in patients with SZ and BD (*Dogan et al., 2018*; *Pruett and Meador-Woodruff, 2020*). These changes have also been observed in the brains of AD patients (*Lehéricy et al., 2007*; *Liguori*

*et al., 2016*; *Liguori et al., 2015*; *Lyros et al., 2020*; *Mullins et al., 2018*; *Paasila et al., 2019*; *Youssef et al., 2018*). However, the observed phenomena are potentially confounded by secondary factors inherent to human studies, such as the administration of antipsychotic drugs (*Halim et al., 2008*). Agonal experiences associated with these disorders may also complicate the interpretation of postmortem study results (*Li et al., 2004*; *Tomita et al., 2004*; *Vawter et al., 2006*). Although some human studies have suggested that medication use is not a major factor in regulating brain pH and lactate levels (*Dager et al., 2004*; *Halim et al., 2008*; *Kato et al., 1998*; *Machado-Vieira et al., 2017*; *Soeiro-de-Souza et al., 2016*), it is technically difficult to exclude the effects of other potential confounding factors in such studies, especially those using postmortem brain samples. Animal models are exempt from such confounding factors and may therefore help confirm whether increased brain lactate levels and decreased pH are involved in the pathophysiology of neuropsychiatric and neuro-degenerative disorders.

Recently, we reported that increased lactate levels and decreased pH are commonly observed in the postmortem brains of five genetic mouse models of SZ, BD, and ASD (*Hagihara et al., 2018*). All mice used in the study were drug-naive, with equivalent agonal states, postmortem intervals, and ages within each strain. An in vivo magnetic resonance spectroscopy (MRS) study showed increased brain lactate levels in another mouse model of SZ (*das Neves Duarte et al., 2012*), suggesting that this change is not a postmortem artifact. Thus, these findings in mouse models suggest that increased lactate levels and decreased pH reflect the underlying pathophysiology of the disorders and are not mere artifacts. However, knowledge of brain lactate and especially pH in animal models of neuro-psychiatric and neurodegenerative diseases is limited to a small number of models. In particular, few studies have examined brain pH and lactate levels in animal models of MDD, epilepsy, and AD. In addition, few studies other than ours *Hagihara et al., 2018* have examined the brain pH and lactate levels in the same samples in relevant animal models. Systematic evaluations using the same platform have not yet been performed. Therefore, given the availability of established relevant animal models, we launched a research project named the 'Brain pH Project' (*Hagihara et al., 2021b*). The aim of this project was to improve our understanding of changes in brain pH, particularly in animal models of neuropsychiatric and neurodegenerative disorders. We have extended our previous small-scale study (*Hagihara et al., 2018*) by using a greater variety of animal models of neuropsychiatric disorders and neurodegenerative disorders, including not only SZ, BD, and ASD, but also MDD, epilepsy, AD, and peripheral diseases or conditions comorbid with psychiatric disorders (e.g. diabetes mellitus [DM], colitis, and peripheral nerve injury). These animal models included 109 strains or conditions of mice, rats, and chicks with genetic modifications, drug treatments, and other experimental manipulations such as exposure to physical and psychological stressors. Of these, 65 strains/conditions of animal models constituted an exploratory cohort (*Hagihara et al., 2021b*) and 44 constituted a confirmatory cohort used to test the hypothesis developed in the initial exploratory studies. We also implemented a statistical learning algorithm that integrated large-scale brain lactate data with comprehensive behavioral measures covering a broad range of behavioral domains (*Takao and Miyakawa, 2006*; e.g. working memory, locomotor activity in a novel environment, sensorimotor gating functions, anxiety-like behavior, and depression-like behavior) to identify behavioral signatures intrinsically related to changes in brain lactate levels. Importantly, by replicating these studies separately in a distinct cohort, we obtained reliable results regarding the potential functional significance of brain lactate changes in animal models of neuropsychiatric disorders.

## Results

### Altered brain pH and lactate levels in animal models of neuropsychiatric and neurodegenerative disorders

The strains/conditions of animals analyzed in this study and the related diseases/conditions are summarized in *Supplementary file 1*. The raw pH and lactate data and detailed information about the animals (age, sex, and storage duration of tissue samples) are shown in *Supplementary file 2*.

Of the 65 strains/conditions in the exploratory cohort, 26 showed significant changes in pH (6 increased, 20 decreased) and 24 showed significant changes in lactate levels (19 increased, 5 decreased) compared with the corresponding control animals ($P<0.05$; *Figure 1A*, *Supplementary file 3*). No strain/condition of animals showed a concomitant significant increase or decrease in pH

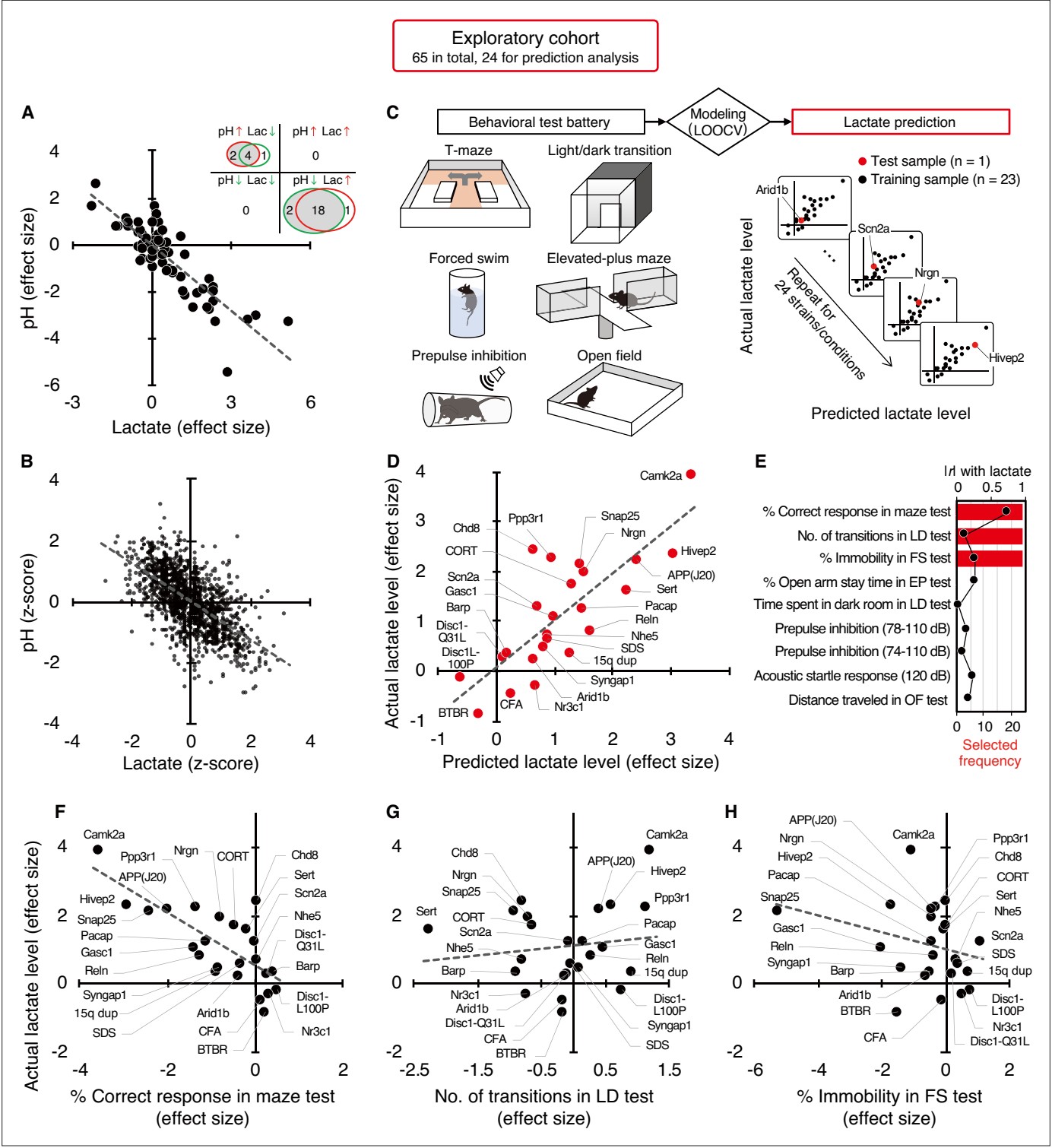

**Figure 1.** Increased brain lactate levels correlated with decreased pH are associated with poor working memory. (**A**) Venn diagrams show the number of strains/conditions of animal models with significant changes (*P*<0.05 compared with the corresponding controls) in brain pH and lactate levels in an exploratory cohort. Scatter plot shows the effect size-based correlations between pH and lactate levels of 65 strains/conditions of animals in the cohort. (**B**) Scatter plot showing the z-score-based correlations between pH and lactate levels of 1,239 animals in the cohort. A z-score was calculated for each animal within the strain/condition and used in this study. (**C**) Schematic diagram of the prediction analysis pipeline. Statistical learning models with leave-one-out cross-validation (LOOCV) were built using a series of behavioral data to predict brain lactate levels in 24 strains/conditions of mice in an exploratory cohort. (**D**) The scatter plot shows significant correlations between predicted and actual lactate levels. (**E**) Feature preference

*Figure 1 continued on next page*

*Figure 1 continued*

for constructing the model to predict brain lactate levels. Bar graphs indicate the selected frequency of behavioral indices in the LOOCV. Line graph indicates absolute correlation coefficient between brain lactate levels and each behavioral measure of the 24 strains/conditions of mice. r, Pearson's correlation coefficient. (F–H) Scatter plot showing correlations between actual brain lactate levels and measures of working memory (correct responses in maze test) (**F**), the number of transitions in the light/dark transition test (**G**), and the percentage of immobility in the forced swim test (**H**).

The online version of this article includes the following figure supplement(s) for figure 1:

**Figure supplement 1.** Normal distribution of effect size values for pH and lactate in the exploratory and confirmatory cohorts.

**Figure supplement 2.** Correlations of brain lactate levels and pH with behavioral measures in an exploratory cohort.

and lactate levels. Effect size-based analysis of 65 strains/conditions showed a significant negative correlation between pH and lactate levels at the strain/condition level ($r=-0.86$, $P=8.45 \times 10^{-20}$; *Figure 1A*, *Figure 1—figure supplement 1*). Furthermore, the Z-score-based meta-analysis of 1,239 animals in the exploratory cohort revealed a highly significant negative correlation between pH and lactate levels at the individual animal level ($r=-0.62$, $P=7.54 \times 10^{-135}$; *Figure 1B*). These results support the idea that decreased brain pH is due to increased lactate levels in pathological conditions associated with neuropsychiatric disorders.

## Poor working memory performance predicts higher brain lactate levels

Most of the animal models analyzed have shown a wide range of behavioral abnormalities, such as deficits in learning and memory, increased depression- and anxiety-like behaviors, and impaired sensorimotor gating. Thereafter, with our comprehensive lactate data, we investigated the potential relationship between lactate changes and behavioral phenotypes in animal models. To this end, we examined whether behavioral patterns could predict brain lactate levels by applying a statistical learning algorithm to reveal intrinsic associations between brain chemical signatures and behavior. Of the 65 animal models, we collected comprehensive behavioral data from 24 mouse models available from public sources (e.g., published papers and database repositories) and in-house studies (*Supplementary file 4*). We constructed an effect-size-based model to predict brain lactate levels from behavioral data using the leave-one-out cross-validation (LOOCV) method (*Figure 1C*, *Supplementary file 5*). Statistical evaluation of the predictive accuracy of the model revealed a significant correlation between the actual and predicted brain lactate levels ($r=0.79$, $P=4.17 \times 10^{-6}$; *Figure 1D*). The calculated root mean square error (RSME) was 0.68. These results indicate that behavioral measures have the potential to predict brain lactate levels in individual models.

Prediction analysis was implemented to evaluate the behavioral measures that were most useful in characterizing the brain lactate levels of the individual strains. The prediction algorithm identified behavioral signatures associated with changes in brain lactate levels by weighting the behavioral measures according to their individual predictive strengths. Thus, we identified behavioral measures associated with changes in brain lactate levels by examining the weighted behavioral measures used for prediction in linear regression. Three out of the nine behavioral measures were selected to build a successful prediction model, and an index of working memory was the top selected measure (*Figure 1E*). According to a simple correlation analysis, working memory measures (correct responses in the maze test) were significantly negatively correlated with brain lactate levels ($r=-0.76$, $P=1.93 \times 10^{-5}$; *Figure 1F*). The other two indices used in the successful prediction model did not show significant correlations with brain lactate levels (number of transitions in the light/dark transition test, $r=0.13$, $P=0.55$, *Figure 1G*; or percentage of immobility in the forced swim test, $r=-0.28$, $P=0.19$, *Figure 1H*). Scatter plots of other behavioral indices are shown in *Figure 1—figure supplement 2*. Behavioral indices with higher correlation coefficients with actual lactate levels were not necessarily preferentially selected to construct the prediction model (*Figure 1E*). These results suggest that higher brain lactate levels are predominantly linearly related to poorer performance in working memory tests in mouse models of neuropsychiatric disorders. Lactate levels had a V-shape-like relationship with the number of transitions in the light/dark transition test (*Figure 1G*) and the percentage of open-arm stay time in the elevated-plus maze test (*Figure 1—figure supplement 2*), which are indices of anxiety-like behavior. This suggests that increased brain lactate levels may also be associated with changes in anxiety-like behaviors, regardless of the direction of the change (increase or decrease).

## Validation studies in an independent confirmatory cohort

In a confirmatory cohort consisting of 44 strains/conditions of animal models, 11 strains/conditions showed significant changes in pH (2 increased, 9 decreased) and 11 in lactate levels (10 increased, 1 decreased) compared with the corresponding controls (P<0.05; *Figure 2A*, *Supplementary file 3*). As observed in the exploratory cohort, there were highly significant negative correlations between brain pH and lactate levels at both the strain/condition ($r$=–0.78, $P$=4.07 × 10$^{-10}$; *Figure 2A*, see *Figure 1— figure supplement 1*) and individual levels ($r$=–0.52, $P$=1.13 × 10$^{-74}$; *Figure 2B*) in this confirmatory cohort.

We then tested the hypothesis developed in the exploratory study that behavioral outcomes predict brain lactate levels. A *priori* power analysis based on an exploratory study ($r$=0.79, *Figure 1D*) estimated that at least 18 strains/conditions of animals would be required to statistically confirm the results at a level of $\alpha$=0.01, $|\rho|$=0.79, 1–$\beta$=0.95 (*Figure 2—figure supplement 1*). Of the 44 strains/ conditions of animals in the confirmatory cohort, we collected comprehensive behavioral data from 27 mouse strains from public sources (e.g. published papers and the Mouse Phenotype Database) and unpublished in-house studies (*Supplementary file 4*) that met the criteria for the aforementioned a priori power analysis. Cross-validation analysis, performed in the same manner as in the exploratory study, showed that behavioral patterns could predict brain lactate levels in the confirmatory cohort ($r$=0.55, p=3.19 × 10$^{-3}$; *Figure 2C and D*). An RMSE value of 0.70 suggests that the prediction accuracy was comparable between the exploratory and confirmatory cohorts (0.68 vs 0.70, respectively). We found that working memory measures (correct responses in the maze test) were the most frequently selected behavioral measures for constructing a successful prediction model (*Figure 2E*), which is consistent with the results of the exploratory study (*Figure 1E*). However, other behavioral measures were selected at different frequencies (*Figure 2E*). Simple correlation analyses showed that working memory measures were negatively correlated with brain lactate levels ($r$=–0.76, p=6.78 × 10$^{-6}$; *Figure 2F*). No significant correlation with lactate levels was found for the acoustic startle response ($r$=–0.26, p=0.21; *Figure 2G*) or the time spent in the dark room in the light/dark transition test ($r$=0.27, p=0.19; *Figure 2H*), which were the second and third behavioral measures selected in the prediction model (*Figure 2E*). Again, behavioral indices with higher correlation coefficients were not necessarily preferentially selected to construct the prediction model (*Figure 2E*, *Figure 2—figure supplement 2*).

## Clustering of 109 strains/conditions of animal models based on changes in brain pH and lactate levels

Combining the exploratory and confirmatory cohorts (109 strains/conditions in total), 37 strains/conditions showed significant changes in pH (8 increased, 29 decreased) and 35 showed significant changes in lactate (29 increased, 6 decreased) compared to the corresponding controls (p<0.05; *Figure 2— figure supplement 3*, *Supplementary file 3*). Highly significant negative correlations were observed between brain pH and lactate levels at both the strain/condition ($r$=–0.80, p=6.99 × 10$^{-26}$; *Figure 2— figure supplement 3A*) and individual levels ($r$=–0.58, p=4.16 × 10$^{-203}$; *Figure 2—figure supplement 3B*), to a greater extent than observed in each cohort. The contribution ratio of lactate to pH, calculated based on the regression coefficient in a linear regression model, was 33.2% at the individual level, suggesting a moderate level of contribution.

In the prediction analysis, behavioral patterns were able to predict brain lactate levels in the combined cohort (51 strains/conditions), as expected (*Figure 2—figure supplement 3C and D*, $r$=0.72, p=3.13 × 10$^{-9}$). Furthermore, behavioral patterns predicted brain pH (*Figure 2—figure supplement 3F and G*, $r$=0.62, p=9.92 × 10$^{-7}$). In both the lactate and pH prediction models, working memory measures were among the most weighted predictors (*Figure 2—figure supplement 3E and H*). Working memory measures were significantly negatively correlated with brain lactate levels and positively correlated with pH (*Figure 2—figure supplement 4*). Moreover, the number of transitions in the light/dark transition test and the percentage of open arm stay time in the elevated-plus maze test showed a V-shape-like relationship with lactate levels (*Figure 2—figure supplement 4*).

Hierarchical clustering based on effect size roughly classified all 109 strains/conditions of animals into four groups: low pH/high lactate group, high pH/low lactate group, moderate-high pH/moderate-low lactate group, and a group with minimal to no changes in pH or lactate. These groups consisted of 30, 2, 15, and 62 strains/conditions of animals, respectively (*Figure 2—figure supplement 5*),

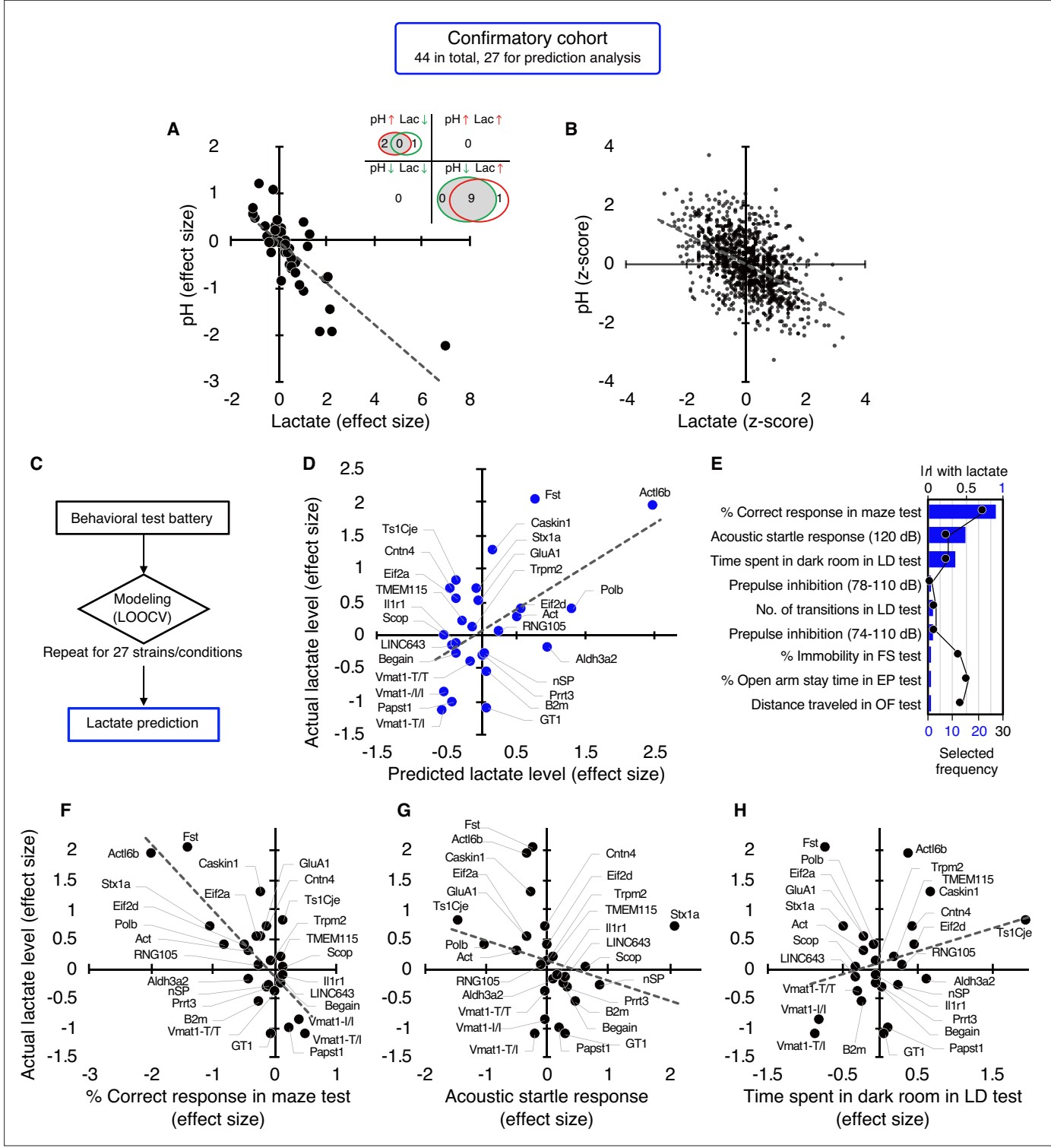

**Figure 2.** Studies in an independent confirmatory cohort validate the negative correlation of brain lactate levels with pH and the association of increased lactate with poor working memory. (**A**) Venn diagrams show the number of strains/conditions of animal models with significant changes (*P*<0.05 compared with the corresponding controls) in brain pH and lactate levels in a confirmatory cohort. Scatter plot shows the effect size-based correlations between pH and lactate levels of 44 strains/conditions of animals in the cohort. (**B**) Scatter plot showing the z-score-based correlations between pH and lactate levels of 1,055 animals in the cohort. (**C**) Statistical learning models with leave-one-out cross-validation (LOOCV) were built using a series of behavioral data to predict brain lactate levels in 27 strains/conditions of mice in the confirmatory cohort. (**D**) The scatter plot shows significant correlations between predicted and actual lactate levels. (**E**) Feature preference for constructing the model to predict brain lactate levels. Bar

*Figure 2 continued on next page*

*Figure 2 continued*

graphs indicate the selected frequency of behavioral indices in the LOOCV. Line graph indicates absolute correlation coefficient between brain lactate levels and each behavioral index of the 27 strains of mice. *r*, Pearson's correlation coefficient. (F–H) Scatter plots showing correlations between actual brain lactate levels and working memory measures (correct responses in the maze test) (**F**), the acoustic startle response at 120 dB (**G**), and the time spent in dark room in the light/dark transition test (**H**). Figure supplements.

The online version of this article includes the following figure supplement(s) for figure 2:

**Figure supplement 1.** A *priori* power analysis to estimate the optimum sample size for the confirmatory experiment.

**Figure supplement 2.** Correlations of brain lactate levels and pH with behavioral measures in a confirmatory cohort.

**Figure supplement 3.** Correlation of increased brain lactate levels and decreased pH and their associations with poor working memory: studies in a combined cohort.

**Figure supplement 4.** Correlations of brain lactate levels and pH with behavioral measures in a combined cohort.

**Figure supplement 5.** Hierarchical clustering of 109 strains/conditions of animals with respect to brain pH and lactate levels.

**Figure supplement 6.** Effects of age, sex, and storage duration on brain pH and lactate levels.

where 'high' and 'low' indicate higher and lower pH and lactate levels in the mutant/experimental animals relative to the corresponding wild-type/control animals, respectively. For example, the low pH/high lactate group included SZ model Nrgn KO mice, SZ/intellectual disability (ID) models Ppp3r1 KO mice and Hivep2 (also known as Shn2) KO mice, AD model APP-J20 Tg mice, ASD model Chd8 KO mice, and social defeat stress-induced depression model mice. Chicks exposed to isolation stress showed decreased brain pH and were included in this group, suggesting that changes in brain pH in response to stress are an interspecies phenomenon. The high pH/low lactate group and moderate-high pH/moderate-low lactate group included mouse models of ASD or developmental delay, such as Shank2 KO, Fmr1 KO, BTBR, Stxbp1 KO, Dyrk1 KO, Auts2 KO, and 15q dup mice (*Supplementary file 1*, *Figure 2—figure supplement 5*).

## Effects of age, sex, and storage duration on brain pH and lactate levels

There was variation among the strains/conditions of the animal models studied with respect to age at sampling, sex, and storage duration of the tissues in the freezer prior to measurements (*Supplementary file 2*). We tested the potential effects of these three factors on the brain pH and lactate levels in samples from wild-type and control rodents. Multivariate linear regression analysis using raw pH values showed that storage duration, but not age or sex, was a significant covariate of brain pH (*Figure 2—figure supplement 6A*). None of these three factors covaried with the raw lactate values. Raw pH values were significantly positively correlated with storage duration (*r*=0.11, p=0.00060; *Figure 2—figure supplement 6D*) but not with age (*r*=0.038, p=0.22; *Figure 2—figure supplement 6B*). No significant correlation was observed between the raw lactate values and age (*r*=0.036, p=0.24) or storage duration (*r*=0.034, p=0.29) (*Figure 2—figure supplement 6C and E*). There were no significant differences in pH (p=0.42) or lactate values (p=0.22) between female and male rodents (*Figure 2—figure supplement 6F and G*).

## Discussion

We performed a large-scale analysis of brain pH and lactate levels in 109 animal models of neuropsychiatric disorders, which revealed the diversity of brain energy metabolism among these animal models. The key findings of this study are as follows: (1) the generality of changes in brain pH and lactate levels across a diverse range of disease models and (2) the association of these phenomenon with specific behaviors. First, this large-scale animal model study revealed that alterations in brain pH/lactate levels can be found in approximately 30% of the animal models examined. This generality suggests a common basis in the neuropathophysiology of not only schizophrenia, bipolar disorder, and ASD, but also of Alzheimer's disease (APP-J20 Tg mice), Down's syndrome (Ts1Cje mice), Mowat–Wilson syndrome (Zeb2 KO mice), Dravet syndrome (Scn1a-A1783V KI mice), tuberous sclerosis complex (Tsc2 KO mice), Ehlers-Danlos syndrome (Tnxb KO mice), and comorbid depression in diabetes (streptozotocin-treated mice) and colitis (dextran sulfate sodium-treated mice). Secondly, this study demonstrated that these phenomenon in the brain are primarily associated with working memory impairment over depression- and anxiety-related behaviors. Importantly, developing these

hypotheses in an exploratory cohort of animals and confirming them in an independent cohort within this study enhances the robustness and reliability of our hypotheses.

Some strains of mice that were considered models of different diseases showed similar patterns of changes in pH and lactate levels. Specifically, the SZ/ID models (Ppp3r1 KO, Nrgn KO mice, and Hivep2 KO mice), BD/ID model (Camk2a KO mice), ASD model (Chd8 KO mice), depression models (mice exposed to social defeat stress, corticosterone-treated mice, and Sert KO mice), and other disease models mentioned above commonly exhibited decreased brain pH and increased lactate levels. BD model Polg1 Tg mice showed no differences in pH or lactate levels. Interestingly, however, other BD model Clock mutant mice and ASD models, such as Shank2 KO (*Lim et al., 2017*), Fmr1 KO, Dyrk1 KO (*Raveau et al., 2018*), Auts2 KO (*Hori et al., 2015*), and 15q dup mice (*Nakatani et al., 2009*), were classified into a group with opposite changes (a group with decreased lactate levels and increased pH). Animal models with different patterns of changes in brain pH and lactate levels may represent subpopulations of patients or specific disease states (*Rossignol and Frye, 2012*). While increased brain lactate levels in neuropsychiatric conditions are almost consistent in the literature, decreased lactate levels have also been found in a cohort of patients with SZ (*Beasley et al., 2009*) and in the euthymic state of BD (*Brady et al., 2012*). Our results from animal studies may also support the idea that patients classified into specific neuropsychiatric disorders based on symptoms are biologically heterogeneous (*Insel and Cuthbert, 2015*) from a brain energy metabolism perspective. Detecting changes in brain pH and lactate levels, whether resulting in an increase or decrease due to their potential bidirectional alterations, using techniques such as MRS may help the diagnosis, subcategorization, and identification of specific disease states of these biologically heterogeneous and spectrum disorders, as has been shown for mitochondrial diseases (*Lin et al., 2003*).

Although previous studies have repeatedly reported that brain pH is decreased in SZ and BD (*Dogan et al., 2018*; *Hagihara et al., 2018*; *Pruett and Meador-Woodruff, 2020*), little is known about brain pH in MDD. Our present study demonstrated that decreased brain pH is a common feature in several preclinical animal models of depression (e.g. mice exposed to social defeat stress, corticosterone-treated mice, and Sert KO mice) and comorbid depression (DM mouse model induced by streptozotocin treatment and colitis mouse model induced by dextran sulfate sodium treatment). These findings raise the possibility that decreased brain pH associated with increased lactate levels may be a common endophenotype in MDD, shared with other neuropsychiatric disorders, and needs to be elucidated in future research.

While we analyzed 109 strains/conditions of animals, we included both those that are widely recognized as animal models for specific neuropsychiatric disorders and those that are not. For example, while interleukin 18 (Il18) KO mice and mitofusin 2 (hMfn2-D210V) Tg mice exhibited changes in pH and lactate levels, the evidence that these genes are associated with specific neuropsychiatric disorders is limited. However, these strains of mice exhibited behavioral abnormalities related to neuropsychiatric disorders, such as depressive-like behaviors and impaired working memory (*Ishikawa et al., 2021*; *Ishikawa et al., 2019*; *Yamanishi et al., 2019*). Furthermore, these mice showed maturation abnormality in the hippocampal dentate gyrus and neuronal degeneration due to mitochondrial dysfunction, respectively, suggesting conceptual validity for utilization as animal models for neuropsychiatric and neurodegenerative disorders (*Burté et al., 2015*; *Cunnane et al., 2020*; *Hagihara et al., 2019*; *Hagihara et al., 2013*). In contrast, mice with heterozygous KO of the synaptic Ras GTPase-activating protein 1 (syngap1), whose mutations have been identified in human patients with ID and ASD, showed an array of behavioral abnormalities relevant to the disorders (*Komiyama et al., 2002*; *Nakajima et al., 2019*), but did not show changes in brain pH or lactate levels. Therefore, while changes in brain pH and lactate levels could be transdiagnostic endophenotypes of neuropsychiatric disorders, they might occur depending on the subpopulation due to the distinct genetic and environmental causes or specific disease states in certain disorders.

The present animal studies revealed a strong negative correlation between brain pH and lactate levels, which supports our previous findings from small-scale animal studies (*Hagihara et al., 2018*). A negative correlation between brain pH and lactate levels was found in a human postmortem study (*Halim et al., 2008*). These results suggest that brain lactate is an important regulator of tissue pH (*Prabakaran et al., 2004*), although we cannot exclude the possibility that other factors, such as neuronal activity-regulated production of carbon dioxide, another metabolic acid, may also contribute to changes in brain pH (*Chesler, 2003*; *Zauner et al., 1995*). Furthermore, the observed pH changes

may be due to the dysregulation of neuronal (*Li et al., 2022*; *Pruett et al., 2023*) and astroglial (*Theparambil et al., 2020*) mechanisms of H⁺ ion transport and buffering to regulate intracellular and extracellular pH homeostasis, which should be investigated in our model animals.

We observed no significant correlation between brain pH and age in the wild-type/control rodents. In human studies, inconsistent results have been obtained regarding the correlation between brain pH and age; some studies showed no significant correlation (*Monoranu et al., 2009*; *Preece and Cairns, 2003*), whereas others showed a negative correlation (*Forester et al., 2010*; *Harrison et al., 1995*). The effect of sex on brain pH has been inconsistent in human studies (*Monoranu et al., 2009*; *Preece and Cairns, 2003*). Systematic analyses focusing on the effects of age and sex on brain pH in animal models may help explain the inconsistency in human studies.

Our prediction analysis revealed that poorer working memory performance in animal models of neuropsychiatric disorders may be predominantly associated with higher lactate levels, which was reliably confirmed in an independent cohort. Higher lactate levels have been associated with lower cognition in individuals with SZ (*Rowland et al., 2016*) and mild cognitive impairment (*Weaver et al., 2015*). Based on these observations, abnormal accumulation of lactate would be expected to have a negative impact on brain function, especially memory formation. However, lactate production stimulated by learning tasks has been suggested to be a requisite for memory formation. Lactate production by astrocytic glycogenolysis and its transport to neurons serves as an energy substrate for neuronal activity and is referred to as astrocyte-neuron lactate shuttle (ANLS). Animal studies have shown that pharmacological disruption of learning task-stimulated lactate production and transport via the ANLS immediately before testing impairs memory formation, as assessed by the plus-shaped maze spontaneous alteration task (testing short-term memory; *Newman et al., 2011*) and inhibitory avoidance task (testing long-term memory; *Descalzi et al., 2019*; *Suzuki et al., 2011*). Collectively, considering that brain lactate levels increase during stimulation in a temporally (and spatially) restricted manner under physiological conditions (*Mangia et al., 2007*; *Schaller et al., 2014*), pathologically sustained elevation of brain lactate levels may have negative effects on brain functions, including memory processing, although causality is unknown. Another possibility is that the reduced consumption of lactate for energy production due to mitochondrial dysfunction in neurons may underlie impaired learning and memory functions in disease conditions. Mitochondrial dysfunction is thought to lead to lactate accumulation due to the insufficient capacity of mitochondrial metabolism to metabolize the lactate produced (*Dogan et al., 2018*; *Regenold et al., 2009*; *Stork and Renshaw, 2005*). Mitochondrial dysfunction has been consistently implicated in several neuropsychiatric disorders, including SZ, BD, MDD, ASD, and AD (*Holper et al., 2019*; *Manji et al., 2012*; *Pei and Wallace, 2018*), among which working memory deficits are a common symptom (*Millan et al., 2012*). In addition, increased lactate levels reflect neuronal activation (*Hagihara et al., 2018*). Thus, activation in brain regions other than the frontal cortex, a brain region critical for working memory (*Andrés, 2003*), interferes with working memory performance, as it has been proposed that the activity of the core brain region may be affected by noise from the rest of the brain during cognitive tasks in patients with SZ (*Foucher et al., 2005*). Moreover, increased lactate may have a positive or beneficial effect on memory function to compensate for its impairment, as lactate administration with an associated increase in brain lactate levels attenuates cognitive deficits in human patients (*Bisri et al., 2016*) and rodent models (*Rice et al., 2002*) of traumatic brain injury. In addition, lactate administration exerts antidepressant effects in a mouse model of depression (*Carrard et al., 2021*; *Carrard et al., 2018*; *Karnib et al., 2019*; *Shaif et al., 2018*). Lactate has also shown to promote learning and memory (*Descalzi et al., 2019*; *Dong et al., 2017*; *El Hayek et al., 2019*; *Lu et al., 2019*; *Roumes et al., 2021*; *Suzuki et al., 2011*), synaptic plasticity (*Herrera-López et al., 2020*; *Yang et al., 2014*; *Zhou et al., 2021*), adult hippocampal neurogenesis (*Lev-Vachnish et al., 2019*), and mitochondrial biogenesis and antioxidant defense (*Akter et al., 2023*), while its effects on adult hippocampal neurogenesis and learning and memory are controversial (*Ikeda et al., 2021*; *Lev-Vachnish et al., 2019*; *Wang et al., 2019*). Moreover, increased lactate levels may also be involved in behavioral changes other than memory deficits, such as anxiety. The results of our previous study showed that increased brain lactate levels were associated with altered anxiety-like behaviors in a social defeat stress model of depression (*Hagihara et al., 2021a*). Further studies are needed to address these hypotheses by chronically inducing deficits in mitochondrial function to manipulate endogenous lactate levels in a brain-region-specific manner and to analyze their effects on working

memory. It is also important to consider whether pH or lactate contributes more significantly to the observed behavioral abnormalities.

There exists a close relationship between neuronal activity and energy metabolism in the brain. In vitro studies have indicated that the uptake of glutamate into astrocytes stimulates glycolysis and lactate production following neuronal excitation (*Pellerin and Magistretti, 1994*). However, an in vivo investigation on cerebellar Purkinje cells has demonstrated that lactate is produced in neurons in an activity-dependent manner, suggesting that astrocytes may not be the sole supplier of lactate to neurons (*Caesar et al., 2008*). Shifts in the neuronal excitation and inhibition (E/I) balance toward excitation of specific neural circuits have been implicated in the pathogenesis and pathophysiology of various neuropsychiatric disorders, including SZ, BD, ASD, AD, and epilepsy (*Brealy et al., 2015*; *Busche and Konnerth, 2016*; *Marín, 2012*; *Nelson and Valakh, 2015*; *Yizhar et al., 2011*). An imbalance favoring excitation could lead to increased energy expenditure and potentially heightened glycolysis. Such alterations in energy metabolism may be associated with increased lactate production. Indeed, in our previous studies using Hivep2 KO mice, characterized by increased brain lactate levels and decreased pH, we observed elevated glutamate levels and upregulated expression of many glycolytic genes in the hippocampus (*Hagihara et al., 2018*; *Takao et al., 2013*). Furthermore, Actl6b (also known as Baf53b) KO mice (*Wenderski et al., 2020*) and APP-J20 Tg mice (*Bomben et al., 2014*; *Brown et al., 2018*; *Palop et al., 2007*) exhibited neuronal hyperexcitation, as evidenced by increased expression of activity-regulated genes and epileptiform discharges recorded by electroencephalography. Dravet syndrome model mice with a clinically relevant SCN1A mutation (Scn1a-A1783V knock-in mice) (*Ricobaraza et al., 2019*) and mutant Snap25 (S187A) knock-in mice (*Kataoka et al., 2011*) developed convulsive seizures. These findings suggest that neuronal hyperexcitation may be one of the common factors leading to increased lactate production and decreased pH in the brain. We consider that alterations in brain pH and lactate levels occur, whether they are a direct and known consequence or indirect and unknown ones of genetic modifications. We have proposed that genetic modifications, along with environmental stimulations, may induce various changes, which subsequently converge toward specific endophenotypes in the brain, such as neuronal hyperexcitation, inflammation, and maturational abnormalities (*Hagihara et al., 2013*; *Yamasaki et al., 2008*). The findings of this study, demonstrating the commonality of alterations in brain pH and lactate levels, align with this concept and suggest that these alterations could serve as brain endophenotypes in multiple neuropsychiatric disorders.

The major limitations of this study include the absence of analyses specific to brain regions or cell types and the lack of functional investigations. Because we used whole brain samples to measure pH and lactate levels, we could not determine whether the observed changes in pH and/or lactate levels occurred ubiquitously throughout the brain or selectively in specific brain region(s) in each strain/condition of the models. It is known that certain molecular expression profiles and signaling pathways display brain region-specific alterations, and in some cases, even exhibit opposing changes in neuropsychiatric disease models (*Floriou-Servou et al., 2018*; *Hosp et al., 2017*; *Reim et al., 2017*). Indeed, brain region-specific increases in lactate levels were observed in human patients with ASD in an MRS study (*Goh et al., 2014*). Furthermore, while increased lactate levels were observed in whole-brain measurements in mice with chronic social defeat stress (*Figure 2—figure supplement 5*; *Hagihara et al., 2021a*), decreased lactate levels were found in the dorsomedial prefrontal cortex (*Yao et al., 2023*). Additionally, it has been reported that the basal intracellular pH differs between neurons and astrocytes (lower in astrocytes than in neurons), and their responsiveness to conditions simulating neural hyperexcitation and the metabolic acidosis in terms of intracellular pH also varies (*Raimondo et al., 2016*; *Salameh et al., 2017*). It would also be possible that the brain region/cell type-specific changes may occur even in animal models in which undetectable changes were observed in the present study. This could be due to the masking of such changes in the analysis when using whole-brain samples. Given the assumption that the brain regions and cell types responsible for pH and lactate changes vary across different strains/conditions, comprehensive studies are needed to thoroughly examine this issue for each animal model individually. This can be achieved through techniques such as evaluating microdissected brain samples, conducting in vivo analyses using pH- or lactate-sensitive biosensor electrodes (*Marunaka et al., 2014*; *Newman et al., 2011*) and MRS (*Davidovic et al., 2011*). Subsequently, based on such findings, it is also necessary to conduct functional

analyses for each model animal by manipulating pH or lactate levels in specific brain regions/cell types and evaluating behavioral phenotypes relevant to neuropsychiatric disorders.

We also note that there are several potential confounding factors in this study. The brain samples analyzed in this study contained cerebral blood. The cerebral blood volume is estimated to be approximately 20–50 µl/g in human and feline brains (*Leenders et al., 1990*; *van Zijl et al., 1998*). When we extrapolate these values to murine brains, it would imply that the proportion of blood contamination in the brain homogenates analyzed is 0.2–0.6%. Additionally, lactate concentrations in the blood are two to three times higher than those in the brains of mice (*Béland-Millar et al., 2017*). Therefore, even if there were differences in the amount of resident blood in the brains between control and experimental animals, the impact of such differences on the lactate measurements would likely be minimal. Other confounding factors include circadian variation and locomotor activity before the brain sampling. Lactate levels are known to exhibit circadian rhythm in the rodent cortex, transitioning gradually from lower levels during the light period to higher levels during the dark period (*Dash et al., 2012*; *Shram et al., 2002*; *Wallace et al., 2020*). The variation in the times of sample collection during the day was basically kept minimized within each strain/condition of animals. However, the sample collection times were not explicitly matched across the different laboratories, which may contribute to variations in the baseline control levels of pH and lactate among different strains/conditions of animals (*Supplementary file 3*). In addition, motor activity and wake/sleep status immediately before brain sampling can also influence brain lactate levels (*Naylor et al., 2012*; *Shram et al., 2002*). These factors have the potential to act as confounding variables in the measurement of brain lactate and pH in animals.

In conclusion, the present study demonstrated that altered brain pH and lactate levels are commonly observed in animal models of SZ, BD, ID, ASD, AD, and other neuropsychiatric disorders. These findings provide further evidence supporting the hypothesis that altered brain pH and lactate levels are not mere artifacts, such as those resulting from medication confounding, but are rather involved in the underlying pathophysiology of some patients with neuropsychiatric disorders. Altered brain energy metabolism or neural hyper- or hypoactivity leading to abnormal lactate levels and pH may serve as a potential therapeutic targets for neuropsychiatric disorders (*Pruett and Meador-Woodruff, 2020*). Future studies are needed to identify effective treatment strategies specific to sets of animal models that could recapitulate the diversity of brain energy metabolism in human disease conditions. This could contribute to the development of treatments for biologically defined subgroups of patients or disease states of debilitating diseases beyond clinically defined boundaries.

# Materials and methods

**Key resources table**

| Reagent type (species) or resource | Designation | Source or reference | Identifiers | Additional information |
|---|---|---|---|---|
| Biological sample (mice, rats, and chicks) | See *Supplementary files 1 and 2* | | | |
| Commercial assay or kit | Lactate Lysing Reagent | Analox Instruments | GMRD-103 | |
| Software, algorithm | EZR software | Saitama Medical Center, Jichi Medical University (*Kanda, 2013*) | | |

## Experimental animals and ethical statement

The animals used in this study are listed in *Supplementary file 1*. Animal experiments were approved by the Institutional Animal Care and Use Committee of Fujita Health University (reference number AP22004) and the relevant committee at each participating institution, based on the Law for the Humane Treatment and Management of Animals and the Standards Relating to the Care and Management of Laboratory Animals and Relief of Pain. Every effort was made to minimize the number of animals used.

## Sample collection

Whole brain samples were collected by one of the following methods:

1. Call for collaborative research worldwide, for example by posting on the website of the relevant scientific society (https://www.ibngs.org/news) and of our institute (http://www.fujita-hu.ac.jp/~cgbb/en/collaborative_research/index.html), and by discussion on a preprint server, *bioRxiv* (https://www.biorxiv.org/content/10.1101/2021.02.02.428362v2).
2. Ask specifically the researchers who have established animal models.
3. Purchase or transfer mouse strains of interest from the repository (e.g. The Jackson Laboratory [https://www.jax.org/], RIKEN BioResource Research Center [https://web.brc.riken.jp/en/]).
4. Rederivation of mouse strains of interest from frozen embryo stocks.

## Sampling and handling of brain samples

We have established a standardized protocol for the sampling and handling of brain samples to minimize potential confounding effects due to technical differences between laboratories and to conduct blinded studies (http://www.fujita-hu.ac.jp/~cgbb/en/collaborative_research/index.html).

## Animals and samples

- Animals: Mice, rats, and other laboratory animals. For genetically engineered animals, mutants and their wild-type littermates should be used.
- Number of animals:>6 per group (identical genetic background, littermate), preferably.
- Sex of animals: All males, all females, or balanced among groups if mixed.
- Samples: Fresh-frozen whole brain.

## Blinded study

The pH and lactate measurements were blinded: Upon sampling, the investigators were supposed to randomize the animals regarding genotype and collect the brain samples in serially numbered tubes. The investigators were asked to provide the genotype information and corresponding serial numbers after the measurements for subsequent statistical analyses.

## Brain sampling procedures

1. Sacrifice the mouse/rat by cervical dislocation followed by decapitation and remove the entire brain from the skull. Do not immerse the brain in buffer solutions or water.
2. Cut the brain along the longitudinal fissure of the cerebrum.
3. Collect the left and right hemispheres in a tube that can be tightly capped like a cryotube and seal the caps with Parafilm (to minimize the effect of carbon dioxide from dry ice on the tissue pH during transport).
4. Quick freeze in liquid nitrogen and store at –80 °C until shipped.
5. Transport the frozen brain on dry ice.

## Measurements of pH and lactate

pH and lactate were measured as previously described (*Hagihara et al., 2018*). Briefly, snap-frozen tissues were homogenized in ice-cold distilled $H_2O$ (5 ml per 500 mg of tissue). The pH of the homogenates was measured using a pH meter (LAQUA F-72, HORIBA, Ltd., Kyoto, Japan) equipped with a Micro ToupH electrode (9618S-10D, HORIBA, Ltd.) after three-point calibration at pH 4.0, pH 7.0, and pH 9.0. The concentration of lactate in the homogenates was determined using a multi-assay analyzer (GM7 MicroStat, Analox Instruments, London, UK) after calibration with 8.0 M lactate standard solution (Lactate Lysing Reagent, GMRD-103, Analox Instruments). A 20 µl aliquot of the centrifuged supernatant (14,000 rpm, 10 min) was used for the measurement.

Effect size (d) was calculated for each strain/condition and each measure (i.e., pH, lactate value, and behavioral index) as followed:

$$d = (M_{mutants} - M_{controls})/S_{pooled}$$

$$S_{pooled} = [(S^2_{mutant} + S^2_{control})/2]^{1/2}$$

The heat map was depicted using the R (version 3.5.2) gplots package.

Z-score transformation, a traditional method of data normalization for direct comparison between different samples and conditions, was applied to each pH or lactate value using individual animal data within each of strain according to the following formula:

$$Z - score = (value_P - mean\ value_{P1...Pn})/standard\ deviation_{P1...Pn}$$

where P is any pH or lactate and P1...Pn represent the aggregate measure of all pH or lactate values.

## Prediction analysis

We collected the comprehensive behavioral data as much as of animal models whose brain pH and lactate levels were examined in this study. We obtained the following behavioral data from 24 animal models in an exploratory cohort from published papers, the Mouse Phenotype Database (http://www.mouse-phenotype.org/) or in-house studies (*Supplementary files 3 and 4*): number of transitions in the light-dark transition test, percentage of immobility in the forced swim test, time spent in open arm in the elevated-plus maze test, prepulse inhibition at 78–110 dB and 74–110 dB, startle response at 120 dB, distance traveled in the open field test, and correct percentage in the T-maze, Y-maze, or eight-arm radial maze test. Literature searches were performed in PubMed and Google Scholar using relevant keywords: name of strain or experimental condition, species (mice or rats), and name of behavioral tests. Among the top hits of the search, data presented as actual values of mean and SD or SEM were used with priority. For some behavioral measures, possible mean and SD values were estimated from the graph presented in the paper. In the matrix of strains/conditions and behavioral measures, those with any missing values were excluded, resulting in nine behavioral measures from 24 strains/conditions of mouse models. The effect size was calculated for each strain/condition and measure and used in the prediction analysis.

Leave-one-out cross-validation was employed to determine whether behavioral measures could predict brain lactate levels for individual mouse strains. From the analyzed behavioral dataset of 24 mouse strains, one sample was selected and excluded to serve as the test data of the cross-validation. Then, a multivariate linear regression model was trained on the remaining 23 samples using a stepwise variable selection procedure with EZR software (version 1.38; Saitama Medical Center, Jichi Medical University, Saitama, Japan) (*Kanda, 2013*), and the test sample was predicted. This was repeated 24 times, with all samples selected once as the test data. Behavioral measures selected at least once in the prediction model were considered as predictive behavioral measures. Prediction performance was analyzed by evaluating the correlation between predicted and actual values for the 24 mouse strains.

For comparability, we performed prediction analyses in a confirmatory cohort using the nine behavioral indices mentioned above, resulting in the inclusion of 27 strains/conditions of animals (*Supplementary files 3 and 4*). In the prediction analyses, the same settings as used in an exploratory cohort were applied to the confirmatory and combined cohorts.

To compare prediction accuracy across cohorts, the root mean squared error (RMSE) was calculated using the following formula:

$$RMSE = [(1/n)\sum_{k=1}^{a}(f_i - y_i)^2]^{1/2}$$

where n is the total number of samples, $f_i$ is the predicted value, and $y_i$ is the actual value.

## Statistical analysis

pH and lactate data were analyzed by unpaired t-test or one-way analysis of variance (ANOVA) or two-way ANOVA followed by post hoc Tukey's multiple comparison test using GraphPad Prism 8 (version 8.4.2; GraphPad Software, San Diego, CA). Correlation analysis was performed using Pearson's correlation coefficient method.

## Acknowledgements

This work was supported by MEXT KAKENHI (Grant No. JP16H06462 to T Miyakawa), MEXT Promotion of Distinctive Joint Research Center Program (Grant No. JPMXP0618217663 to T Miyakawa), JSPS KAKENHI (Grant Nos. JP20H00522 and JP16H06276 (AdAMS) to T Miyakawa; Grant Nos.

JP18K07378 and JP21K19314 to H Hagihara), and AMED Strategic Research Program for Brain Sciences (Grant No. JP18dm0107101 to T Miyakawa).

## Additional information

### Competing interests

Satoshi Yamamoto, Naoya Nishimura: Employee of Takeda Pharmaceutical Company, Ltd. The other authors declare that no competing interests exist.

### Funding

| Funder | Grant reference number | Author |
| --- | --- | --- |
| Ministry of Education, Culture, Sports, Science and Technology | JP16H06462 | Tsuyoshi Miyakawa |
| Ministry of Education, Culture, Sports, Science and Technology | JPMXP0618217663 | Tsuyoshi Miyakawa |
| Japan Society for the Promotion of Science | JP20H00522 | Tsuyoshi Miyakawa |
| Japan Society for the Promotion of Science | JP16H06276 | Tsuyoshi Miyakawa |
| Japan Society for the Promotion of Science | JP18K07378 | Hideo Hagihara |
| Japan Society for the Promotion of Science | JP21K19314 | Hideo Hagihara |
| Japan Agency for Medical Research and Development | JP18dm0107101 | Tsuyoshi Miyakawa |

The funders had no role in study design, data collection and interpretation, or the decision to submit the work for publication.

### Author contributions

Hideo Hagihara, Conceptualization, Formal analysis, Funding acquisition, Investigation, Writing - original draft, Writing – review and editing; Hirotaka Shoji, Satoko Hattori, Masafumi Ihara, Mihiro Shibutani, Izuho Hatada, Kei Hori, Mikio Hoshino, Akito Nakao, Yasuo Mori, Shigeo Okabe, Masayuki Matsushita, Anja Urbach, Yuta Katayama, Akinobu Matsumoto, Keiichi I Nakayama, Shota Katori, Takuya Sato, Takuji Iwasato, Haruko Nakamura, Yoshio Goshima, Matthieu Raveau, Tetsuya Tatsukawa, Kazuhiro Yamakawa, Noriko Takahashi, Haruo Kasai, Johji Inazawa, Ikuo Nobuhisa, Tetsushi Kagawa, Tetsuya Taga, Mohamed Darwish, Hirofumi Nishizono, Keizo Takao, Kiran Sapkota, Kazutoshi Nakazawa, Tsuyoshi Takagi, Haruki Fujisawa, Yoshihisa Sugimura, Kyosuke Yamanishi, Lakshmi Rajagopal, Nanette Deneen Hannah, Herbert Y Meltzer, Tohru Yamamoto, Shuji Wakatsuki, Toshiyuki Araki, Katsuhiko Tabuchi, Tadahiro Numakawa, Hiroshi Kunugi, Freesia L Huang, Atsuko Hayata-Takano, Hitoshi Hashimoto, Kota Tamada, Toru Takumi, Takaoki Kasahara, Tadafumi Kato, Isabella A Graef, Gerald R Crabtree, Nozomi Asaoka, Hikari Hatakama, Shuji Kaneko, Takao Kohno, Mitsuharu Hattori, Yoshio Hoshiba, Ryuhei Miyake, Kisho Obi-Nagata, Akiko Hayashi-Takagi, Léa J Becker, Ipek Yalcin, Yoko Hagino, Hiroko Kotajima-Murakami, Yuki Moriya, Kazutaka Ikeda, Hyopil Kim, Bong-Kiun Kaang, Hikari Otabi, Yuta Yoshida, Atsushi Toyoda, Noboru H Komiyama, Seth GN Grant, Michiru Ida-Eto, Masaaki Narita, Ken-ichi Matsumoto, Emiko Okuda-Ashitaka, Iori Ohmori, Tadayuki Shimada, Kanato Yamagata, Hiroshi Ageta, Kunihiro Tsuchida, Kaoru Inokuchi, Takayuki Sassa, Akio Kihara, Motoaki Fukasawa, Nobuteru Usuda, Tayo Katano, Teruyuki Tanaka, Yoshihiro Yoshihara, Michihiro Igarashi, Takashi Hayashi, Kaori Ishikawa, Satoshi Yamamoto, Naoya Nishimura, Kazuto Nakada, Shinji Hirotsune, Kiyoshi Egawa, Kazuma Higashisaka, Yasuo Tsutsumi, Shoko Nishihara, Noriyuki Sugo, Takeshi Yagi, Naoto Ueno, Tomomi Yamamoto, Yoshihiro Kubo, Rie Ohashi,

Nobuyuki Shiina, Kimiko Shimizu, Sayaka Higo-Yamamoto, Katsutaka Oishi, Hisashi Mori, Tamio Furuse, Masaru Tamura, Hisashi Shirakawa, Daiki X Sato, Yukiko U Inoue, Yuriko Komine, Tetsuo Yamamori, Kenji Sakimura, Resources, Investigation, Writing – review and editing; Giovanni Sala, Investigation, Writing – review and editing; Yoshihiro Takamiya, Mika Tanaka, Investigation; Takayoshi Inoue, Resources, Writing – review and editing, Investigation; Tsuyoshi Miyakawa, Conceptualization, Resources, Funding acquisition, Investigation, Writing - original draft, Writing – review and editing

**Author ORCIDs**
Hideo Hagihara ⓘ http://orcid.org/0000-0001-9602-9518
Mikio Hoshino ⓘ http://orcid.org/0000-0002-9526-3473
Akito Nakao ⓘ http://orcid.org/0000-0002-2296-8245
Shigeo Okabe ⓘ http://orcid.org/0000-0003-1216-8890
Kazuhiro Yamakawa ⓘ http://orcid.org/0000-0002-1478-4390
Haruo Kasai ⓘ http://orcid.org/0000-0003-2327-9027
Keizo Takao ⓘ http://orcid.org/0000-0002-4734-3583
Kyosuke Yamanishi ⓘ http://orcid.org/0000-0001-5130-8193
Tohru Yamamoto ⓘ https://orcid.org/0000-0002-1652-0233
Hitoshi Hashimoto ⓘ https://orcid.org/0000-0001-6548-4016
Kota Tamada ⓘ https://orcid.org/0000-0002-6575-6189
Toru Takumi ⓘ http://orcid.org/0000-0001-7153-266X
Takaoki Kasahara ⓘ http://orcid.org/0000-0002-0953-5146
Takao Kohno ⓘ http://orcid.org/0000-0003-2557-4281
Bong-Kiun Kaang ⓘ http://orcid.org/0000-0001-7593-9707
Seth GN Grant ⓘ http://orcid.org/0000-0001-8732-8735
Tadayuki Shimada ⓘ http://orcid.org/0000-0001-7333-9847
Kunihiro Tsuchida ⓘ http://orcid.org/0000-0002-3983-5756
Kaoru Inokuchi ⓘ http://orcid.org/0000-0002-5393-3133
Takayuki Sassa ⓘ http://orcid.org/0000-0003-3145-9829
Akio Kihara ⓘ http://orcid.org/0000-0001-5889-0788
Michihiro Igarashi ⓘ http://orcid.org/0000-0003-1474-3385
Takashi Hayashi ⓘ http://orcid.org/0000-0003-3591-2109
Kaori Ishikawa ⓘ http://orcid.org/0000-0001-9895-5996
Shoko Nishihara ⓘ http://orcid.org/0000-0002-1668-2603
Naoto Ueno ⓘ http://orcid.org/0000-0002-8375-2317
Yoshihiro Kubo ⓘ http://orcid.org/0000-0001-6707-0837
Nobuyuki Shiina ⓘ http://orcid.org/0000-0002-1854-4239
Kimiko Shimizu ⓘ http://orcid.org/0000-0003-4943-2552
Katsutaka Oishi ⓘ http://orcid.org/0000-0002-4870-0837
Hisashi Mori ⓘ http://orcid.org/0000-0001-9743-2456
Hisashi Shirakawa ⓘ http://orcid.org/0000-0002-4129-0978
Daiki X Sato ⓘ https://orcid.org/0000-0002-9527-8253
Yukiko U Inoue ⓘ https://orcid.org/0000-0002-4105-9127
Yuriko Komine ⓘ http://orcid.org/0000-0002-0097-8813
Kenji Sakimura ⓘ http://orcid.org/0000-0002-8091-8879
Tsuyoshi Miyakawa ⓘ http://orcid.org/0000-0003-0137-8200

**Ethics**

Animal experiments were approved by the Institutional Animal Care and Use Committee of Fujita Health University and the relevant committee at each participating institution, based on the Law for the Humane Treatment and Management of Animals and the Standards Relating to the Care and Management of Laboratory Animals and Relief of Pain. Every effort was made to minimize the number of animals used. (reference number AP22004).

Reviewer #1 (Public Review): https://doi.org/10.7554/eLife.89376.3.sa1
Reviewer #2 (Public Review): https://doi.org/10.7554/eLife.89376.3.sa2
Author Response https://doi.org/10.7554/eLife.89376.3.sa3

# Additional files

## Supplementary files

- Supplementary file 1. Animal models used in this study.
- Supplementary file 2. Raw data of brain pH and lactate, as well as information about animals and brain samples (age at sampling, sex, duration of storage in the freezer, and treatment procedures).
- Supplementary file 3. Detailed statistical analysis of pH and lactate measurements in 109 strains/conditions of animals.
- Supplementary file 4. Source of behavioral data used in prediction analysis.
- Supplementary file 5. The effect size values used in prediction analysis.
- MDAR checklist

## Data availability

The data analyzed in this study are available in *Supplementary files 1–5*.

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
