## [Editor Report · eLife assessment]

The manuscript offers **useful** descriptive insights into the potential influence of whole-brain lactate and pH levels on the manifestation of behavioral phenotypes seen in diverse animal models of neuropsychiatric disorders. However, reviewers have raised concerns about the potential loss of specificity in capturing regional and cell-type-specific effects when relying solely on whole-brain analysis methods. While the evidence supporting the conclusions is largely **solid**, the robustness of these conclusions could be enhanced by the inclusion of additional data and further analysis.

---

## [Referee Report · Reviewer #1 (Public Review)]

Summary

In this manuscript, Hagihara et al. characterized the relationship between the changes in lactate and pH and the behavioral phenotypes in different animal models of neuropsychiatric disorders at a large-scale level. The authors have previously reported that increased lactate levels and decreased pH are commonly observed in the brains of five genetic mouse models of schizophrenia (SZ), bipolar disorder (BD), and autism spectrum disorder (ASD). In this study, they expanded the detection range to 109 strains or conditions of animal models, covering neuropsychiatric disorders and neurodegenerative disorders. Through statistical analysis of the first 65 strains/conditions of animal models which were set as exploratory cohort, the authors found that most strains showed decreased pH and increased lactate levels in the brains. There was a significant negative correlation between pH and lactate levels both at the strain/condition level and the individual animal level. Besides, only working memory was negatively correlated with brain lactate levels. These results were successfully duplicated by studying the confirmative cohort, including 44 strains/conditions of animal models. In all strains/conditions, the lactate levels were not correlated with age, sex, or storage duration of brain samples.

Strengths

1. The manuscript is well-written and structured. In particular, the discussion is really nice, covering many potential mechanisms for the altered lactate levels in these disease models.

2. Tremendous efforts were made to recruit a huge number of various animal models, giving the conclusions sufficient power.

Weaknesses

1. The biggest concern of this study is the limited novelty. The point of "altered pH and/or lactate levels in the brains from human and rodent animals of neuropsychiatric disorders" has been reported by the same lab and other groups in many previous papers.

2. This study is mostly descriptive, lacking functional investigations. Although a larger cohort of animal models were studied which makes the conclusion more solid, limited conceptual advance is contributed to the relevant field, as we are still not clear about what the altered levels of pH and lactate mean for the pathogenesis of neuropsychiatric disorders.

3. The experiment procedure is also a concern. The brains from animal models were acutely collected without cardiac perfusion in this study, which suggests that resident blood may contaminate the brain samples. The lactate is enriched in the blood, making it a potential confounded factor to affect the lactate levels as well as pH in the brain samples.

4. The lactate and pH levels may also be affected by other confounded factors, such as circadian period, and locomotor activity before the mice were sacrificed. This should also be discussed in the paper.

5. Another concern is the animal models. Although previous studies have demonstrated that dysfunctions of these genes could cause related phenotypes for certain disorders, many of them are not acknowledged by the field as reliable disease models. Besides, gene deficiency could also cause many known or unknown unrelated phenotypes, which may contribute to the altered levels of lactate and pH, too. In this circumstance, the conclusion "pH and lactate levels are transdiagnostic endophenotype of neuropsychiatric disorders" is somewhat overstated.

6. The negative correlationship between pH and lactate is rather convincing. However, how much the contribution of lactate to pH is not tested. In addition, regarding pH and lactate, which factor contributes most to the pathogenesis of neuropsychiatric disorders is also unclear. These questions may need to be addressed in the future study.

7. The authorship is open to question. Most authors listed in this paper may only provide mice strains or brain samples. Maybe it is better just to acknowledge them in the acknowledgements section.

8. The last concern is about the significance of this study. Although the majority of strains showed increased lactate, some still showed decreased lactate levels in the brains. These results suggested that lactate or pH is an endophenotype for neuropsychiatric disorders, but it is hard to serve as a good diagnostic index as the change is not unidirectional in different disorders. In other words, the relationship between lactate level and neuropsychiatric disorders is not exclusive.

---

## [Referee Report · Reviewer #2 (Public Review)]

Hagihara et al. conducted a study investigating the correlation between decreased brain pH, increased brain lactate, and poor working memory. They found altered brain pH and lactate levels in animal models of neuropsychiatric and neurodegenerative disorders. Their study suggests that poor working memory performance may predict higher brain lactate levels.

However, the study has some significant limitations. One major concern is that the authors examined whole-brain pH and lactate levels, which might not fully represent the complexity of disease states. Different brain regions and cell types may have distinct protein and metabolite profiles, leading to diverse disease outcomes. For instance, certain brain regions like the hippocampus and nucleus accumbens exhibit opposite protein/signaling pathways in neuropsychiatric disease models.

Moreover, the memory tests used in the study are specific to certain brain regions, but the authors did not measure lactate levels in those regions. Without making lactate measurements in brain-regions and cell types involved in these diseases, any conclusions regarding the role of lactate in CNS diseases is premature.

Additionally, evidence suggests that exogenous treatment with lactate has positive effects, such as antidepressant effects in multiple disease models (Carrard et al., 2018, Carrard et al., 2021, Karnib et al., 2019, Shaif et al., 2018). It also promotes learning, memory formation, neurogenesis, and synaptic plasticity (Suzuki et al., 2011, Yang et al., 2014, Weitian et al., 2015, Dong et al., 2017, El Hayek et al. 2019, Wang et al., 2019, Lu et al., 2019, Lev-Vachnish et a.l, 2019, Descalzi G et al., 2019, Herrera-López et al., 2020, Ikeda et al., 2021, Zhou et al., 2021,Roumes et al., 2021, Frame et al., 2023, Akter et al., 2023).

In conclusion, the relevance of total brain pH and lactate levels as indicators of the observed correlations is controversial, and evidence points towards lactate having more positive rather than negative effects. It is important that the authors perform studies looking at brain-region-specific concentrations of lactate and that they modulate lactate levels (decrease) in animal models of disease to validate their conclusions. It is also important to consider the above-mentioned studies before concluding that "altered brain pH and lactate levels are rather involved in the underlying pathophysiology of some patients with neuropsychiatric disorders" and that "lactate can serve as a potential therapeutic target for neuropsychiatric disorders".

---

## [Author Response]

The following is the authors’ response to the original reviews.

**Public Reviews:**

**Reviewer #1 (Public Review):**
SummaryIn this manuscript, Hagihara et al. characterized the relationship between the changes in lactate and pH and the behavioral phenotypes in different animal models of neuropsychiatric disorders at a large-scale level. The authors have previously reported that increased lactate levels and decreased pH are commonly observed in the brains of five genetic mouse models of schizophrenia (SZ), bipolar disorder (BD), and autism spectrum disorder (ASD). In this study, they expanded the detection range to 109 strains or conditions of animal models, covering neuropsychiatric disorders and neurodegenerative disorders. Through statistical analysis of the first 65 strains/conditions of animal models which were set as exploratory cohort, the authors found that most strains showed decreased pH and increased lactate levels in the brains. There was a significant negative correlation between pH and lactate levels both at the strain/condition level and the individual animal level. Besides, only working memory was negatively correlated with brain lactate levels. These results were successfully duplicated by studying the confirmative cohort, including 44 strains/conditions of animal models. In all strains/conditions, the lactate levels were not correlated with age, sex, or storage duration of brain samples.Strengths1. The manuscript is well-written and structured. In particular, the discussion is really nice, covering many potential mechanisms for the altered lactate levels in these disease models.1. Tremendous efforts were made to recruit a huge number of various animal models, giving the conclusions sufficient power.

We are grateful to Reviewer #1 for the positive evaluation of our manuscript. As indicated in the responses that follow, we have taken all the comments and suggestions made by the reviewer into account in the revised version of our paper.

Weaknesses1. The biggest concern of this study is the limited novelty. The point of "altered pH and/or lactate levels in the brains from human and rodent animals of neuropsychiatric disorders" has been reported by the same lab and other groups in many previous papers.

The previous study mentioned by the reviewer evaluated a small number of animal models of psychiatric disorders. The novelty of this study is underscored by two key findings: (1) the generality of changes in brain pH and lactate levels across a diverse range of disease models, and (2) the association of these phenomenon with specific behaviors. First, this large-scale animal model study revealed that alterations in brain pH/lactate levels can be found in approximately 30% of the animal models examined. This generality suggests a common basis in the neuropathophysiology of not only schizophrenia, bipolar disorder, and ASD, but also of Alzheimer’s disease (APP-J20 Tg mice), Down’s syndrome (Ts1Cje mice), Mowat–Wilson syndrome (Zeb2 KO mice), Dravet syndrome (Scn1a-A1783V KI mice), tuberous sclerosis complex (Tsc2 KO mice), Ehlers-Danlos syndrome (Tnxb KO mice), and comorbid depression in diabetes (streptozotocin-treated mice) and colitis (dextran sulfate sodium-treated mice). Secondly, this study demonstrated that these phenomenon in the brain are primarily associated with working memory impairment over depression- and anxiety-related behaviors. Importantly, developing these hypotheses in an exploratory cohort of animals and confirming them in an independent cohort within this study enhances the robustness and reliability of our hypotheses, which we believe are equally crucial as their novelty. Accordingly, we have revised the discussion section as follows (page 31, line 7):

Original text

"We performed a large-scale analysis of brain pH and lactate levels in 109 animal models of neuropsychiatric disorders, which revealed the diversity of brain energy metabolism among these animal models. Some strains of mice that were considered models of different diseases showed similar patterns of changes in pH and lactate levels. Specifically, the SZ/ID models (Ppp3r1 KO, Nrgn KO mice, and Hivep2 KO mice), BD/ID model (Camk2a KO mice), ASD model (Chd8 KO mice), depression models (mice exposed to social defeat stress, corticosterone-treated mice, and Sert KO mice), AD model (APP-J20 Tg mice), and DM model (Il18 KO and STZ-treated mice) commonly exhibited decreased brain pH and increased lactate levels."

Revised text

"We performed a large-scale analysis of brain pH and lactate levels in 109 animal models of neuropsychiatric disorders, which revealed the diversity of brain energy metabolism among these animal models. The key findings of this study are as follows: (1) the generality of changes in brain pH and lactate levels across a diverse range of disease models, and (2) the association of these phenomenon with specific behaviors. First, this large-scale animal model study revealed that alterations in brain pH/lactate levels can be found in approximately 30% of the animal models examined. This generality suggests a common basis in the neuropathophysiology of not only schizophrenia, bipolar disorder, and ASD, but also of Alzheimer’s disease (APP-J20 Tg mice), Down’s syndrome (Ts1Cje mice), Mowat–Wilson syndrome (Zeb2 KO mice), Dravet syndrome (Scn1a-A1783V KI mice), tuberous sclerosis complex (Tsc2 KO mice), Ehlers-Danlos syndrome (Tnxb KO mice), and comorbid depression in diabetes (streptozotocin-treated mice) and colitis (dextran sulfate sodium-treated mice). Secondly, this study demonstrated that these phenomenon in the brain are primarily associated with working memory impairment over depression- and anxiety-related behaviors. Importantly, developing these hypotheses in an exploratory cohort of animals and confirming them in an independent cohort within this study enhances the robustness and reliability of our hypotheses."

1. This study is mostly descriptive, lacking functional investigations. Although a larger cohort of animal models were studied which makes the conclusion more solid, limited conceptual advance is contributed to the relevant field, as we are still not clear about what the altered levels of pH and lactate mean for the pathogenesis of neuropsychiatric disorders.

We agree with the reviewer’s comment. To address this issue, it is necessary to comprehensively identify brain regions and cell types responsible for pH and lactate changes in each strain/condition of animals, as these may differ among them. Subsequently, based on such findings, we can then proceed with functional investigations that specifically target the identified brain regions/cell types. However, conducting such investigations would require a significant amount of time to complete, approximately 2–3 years, and is beyond the scope of this study. Therefore, we would like to conduct such studies in the future. We have mentioned this limitation by revising the discussion section of this study as follows (page 43, line 5):

Original text

"Because we used whole brain samples to measure pH and lactate levels, we could not determine whether the observed changes in pH and/or lactate levels occurred ubiquitously throughout the brain or selectively in specific brain region(s) in each strain/condition of the models. Indeed, brain region-specific increases in lactate levels were observed in human patients with ASD in an MRS study (Goh et al., 2014). Furthermore, while increased lactate levels were observed in whole-brain measurements in mice with chronic social defeat stress (Figure S7) (Hagihara et al., 2021a), decreased lactate levels were found in the dorsomedial prefrontal cortex (Yao et al., 2023). The brain region-specific changes may occur even in animal models in which undetectable changes were observed in the present study. This could be due to the masking of such changes in the analysis when using whole-brain samples. Further studies are needed to address this issue by measuring microdissected brain samples and performing in vivo analyses using pH- or lactate-sensitive biosensor electrodes (Marunaka et al., 2014; Newman et al., 2011) and MRS (Davidovic et al., 2011)."

Revised text:

"The major limitations of this study include the absence of analyses specific to brain regions or cell types and the lack of functional investigations. Because we used whole brain samples to measure pH and lactate levels, we could not determine whether the observed changes in pH and/or lactate levels occurred ubiquitously throughout the brain or selectively in specific brain region(s) in each strain/condition of the models. It is known that certain molecular expression profiles and signaling pathways display brain region-specific alterations, and in some cases, even exhibit opposing changes in neuropsychiatric disease models (Hosp et al., 2017; Floriou-Servou et al. 2018; Reim et al., 2017). Indeed, brain region-specific increases in lactate levels were observed in human patients with ASD in an MRS study (Goh et al., 2014). Furthermore, while increased lactate levels were observed in whole-brain measurements in mice with chronic social defeat stress (Figure S7) (Hagihara et al., 2021a), decreased lactate levels were found in the dorsomedial prefrontal cortex (Yao et al., 2023). Additionally, it has been reported that the basal intracellular pH differs between neurons and astrocytes (lower in astrocytes than in neurons), and their responsiveness to conditions simulating neural hyperexcitation and the metabolic acidosis in terms of intracellular pH also varies (Raimondo et al., 2016; Salameh et al., 2017). It would also be possible that the brain region/cell type-specific changes may occur even in animal models in which undetectable changes were observed in the present study. This could be due to the masking of such changes in the analysis when using whole-brain samples. Given the assumption that the brain regions and cell types responsible for pH and lactate changes vary across different strains/conditions, comprehensive studies are needed to thoroughly examine this issue for each animal model individually. This can be achieved through techniques such as evaluating microdissected brain samples, conducting in vivo analyses using pH- or lactate-sensitive biosensor electrodes (Marunaka et al., 2014; Newman et al., 2011), and MRS (Davidovic et al., 2011). Subsequently, based on such findings, it is also necessary to conduct functional analyses for each model animal by manipulating pH or lactate levels in specific brain regions/cell types and evaluating behavioral phenotypes relevant to neuropsychiatric disorders."

1. The experiment procedure is also a concern. The brains from animal models were acutely collected without cardiac perfusion in this study, which suggests that resident blood may contaminate the brain samples. The lactate is enriched in the blood, making it a potential confounded factor to affect the lactate levels as well as pH in the brain samples.

We thank the reviewer for pointing this out. We have discussed this issue as follows (page 45, line 4):

We also note that there are several potential confounding factors in this study. The brain samples analyzed in this study contained cerebral blood. The cerebral blood volume is estimated to be approximately 20–50 μl/g in human and feline brains (Leenders et al., 1990; van Zijl et al., 1998). When we extrapolate these values to murine brains, it would imply that the proportion of blood contamination in the brain homogenates analyzed is 0.2–0.6%. Additionally, lactate concentrations in the blood are two to three times higher than those in the brains of mice (Béland-Millar et al., 2017). Therefore, even if there were differences in the amount of resident blood in the brains between control and experimental animals, the impact of such differences on the lactate measurements would likely be minimal.

1. The lactate and pH levels may also be affected by other confounded factors, such as circadian period, and locomotor activity before the mice were sacrificed. This should also be discussed in the paper.

Following the reviewer’s suggestion, we have discussed the matter as follows (page 45, line 12):Other confounding factors include circadian variation and locomotor activity before the brain sampling. Lactate levels are known to exhibit circadian rhythm in the rodent cortex, transitioning gradually from lower levels during the light period to higher levels during the dark period (Dash et al., 2012; Shram et al., 2002; Wallace et al., 2022). The variation in the times of sample collection during the day was basically kept minimized within each strain/condition of animals. However, the sample collection times were not explicitly matched across the different laboratories, which may contribute to variations in the baseline control levels of pH and lactate among different strains/conditions of animals (Table S3). In addition, motor activity and wake/sleep status immediately before brain sampling can also influence brain lactate levels (Neylor et al., 2012; Shram et al., 2002). These factors have the potential to act as confounding variables in the measurement of brain lactate and pH in animals.

1. Another concern is the animal models. Although previous studies have demonstrated that dysfunctions of these genes could cause related phenotypes for certain disorders, many of them are not acknowledged by the field as reliable disease models. Besides, gene deficiency could also cause many known or unknown unrelated phenotypes, which may contribute to the altered levels of lactate and pH, too. In this circumstance, the conclusion "pH and lactate levels are transdiagnostic endophenotype of neuropsychiatric disorders" is somewhat overstated.

We thank the reviewer for pointing this out. We should have taken this issue into consideration. Accordingly, we have discussed this issue as the limitation of this study in the discussion section as follows (page 34, line 14):

"While we analyzed 109 strains/conditions of animals, we included both those that are widely recognized as animal models for specific neuropsychiatric disorders and those that are not. For example, while interleukin 18 (Il18) KO mice and mitofusin 2 (hMfn2-D210V) Tg mice exhibited changes in pH and lactate levels, the evidence that these genes are associated with specific neuropsychiatric disorders is limited. However, these strains of mice exhibited behavioral abnormalities related to neuropsychiatric disorders, such as depressive-like behaviors and impaired working memory (Ishikawa et al., 2019, 2021; Yamanishi et al., 2019). Furthermore, these mice showed maturation abnormality in the hippocampal dentate gyrus and neuronal degeneration due to mitochondrial dysfunction, respectively, suggesting conceptual validity for utilization as animal models for neuropsychiatric and neurodegenerative disorders (Cunnane, et al., 2021; Burté et al., 2015; Hagihara et al., 2013, 2019). In contrast, mice with heterozygous KO of the synaptic Ras GTPase-activating protein 1 (syngap1), whose mutations have been identified in human patients with ID and ASD, showed an array of behavioral abnormalities relevant to the disorders (Komiyama et al., 2002; Nakajima et al., 2019), but did not show changes in brain pH or lactate levels. Therefore, while changes in brain pH and lactate levels could be transdiagnostic endophenotypes of neuropsychiatric disorders, they might occur depending on the subpopulation due to the distinct genetic and environmental causes or specific disease states in certain disorders."

Regarding the latter point suggested by the reviewer, we consider that alterations in brain pH and lactate levels occur, whether they are a direct and known consequence or indirect and unknown ones of genetic modifications. We have proposed that genetic modifications, along with environmental stimulations, may induce various changes, which subsequently converge toward specific endophenotypes in the brain, such as neuronal hyperexcitation, inflammation, and maturational abnormalities (Hagihara et al., 2013; Yamasaki et al., 2008). The findings of this study, demonstrating the commonality of alteration of brain pH and lactate levels, align with this concept, suggesting that these alterations could serve as brain endophenotypes in multiple neuropsychiatric disorders. We have revised the discussion section as follows (page 42, line 8):

Original text

"These findings suggest that the observed increase in lactate production and subsequent decrease in pH in whole-brain samples may be attributed to the hyperactivity of specific neural circuits in a subset of the examined animal models."

Revised text

"These findings suggest that neuronal hyperexcitation may be one of the common factors leading to increased lactate production and decreased pH in the brain. We consider that alterations in brain pH and lactate levels occur, whether they are a direct and known consequence or indirect and unknown ones of genetic modifications. We have proposed that genetic modifications, along with environmental stimulations, may induce various changes, which subsequently converge toward specific endophenotypes in the brain, such as neuronal hyperexcitation, inflammation, and maturational abnormalities (Hagihara et al., 2013; Yamasaki et al., 2008). The findings of this study, demonstrating the commonality of alterations in brain pH and lactate levels, align with this concept and suggest that these alterations could serve as brain endophenotypes in multiple neuropsychiatric disorders."

1. The negative correlationship between pH and lactate is rather convincing. However, how much the contribution of lactate to pH is not tested. In addition, regarding pH and lactate, which factor contributes most to the pathogenesis of neuropsychiatric disorders is also unclear. These questions may need to be addressed in the future study.

To estimate the degree of contribution of lactate to pH, we determined the contribution ratio using the regression coefficient within a linear regression model applied to a combined cohort. The results showed that 33.2% of changes in pH may be explained by changes in lactate level. We have added the following text in the Results section (page 28, line 7).

The contribution ratio of lactate to pH, calculated based on the regression coefficient in a linear regression model, was 33.2% at the individual level, suggesting a moderate level of contribution.

Regarding the latter suggestion, we would like to address the issue in the future study. Accordingly, we have added the following sentence in the discussion section (page 40, line 11):

Original text

"Further studies are needed to address these hypotheses by chronically inducing deficits in mitochondrial function to manipulate endogenous lactate levels in a brain region-specific manner and to analyze their effects on working memory."

Revised text

"Further studies are needed to address these hypotheses by chronically inducing deficits in mitochondrial function to manipulate endogenous lactate levels in a brain region-specific manner and to analyze their effects on working memory. It is also important to consider whether pH or lactate contributes more significantly to the observed behavioral abnormalities."

1. The authorship is open to question. Most authors listed in this paper may only provide mice strains or brain samples. Maybe it is better just to acknowledge them in the acknowledgments section.

In the light of the current circumstances, wherein there is no universally agreed definition of authorship (the Committee on Publication Ethics1), we acknowledge the reviewer’s concern. Collecting a comprehensive range of mouse strains and brain samples is a fundamental principle of this study. Maintaining mouse lines, breeding mice, genotyping, drug administration, and preparation of brain samples each require specialized expertise. Therefore, the scientific and technical contributions of individuals who only provided mouse strains or brain samples was also crucial for obtaining the data essential to this study. In accordance with the authorship guidelines outlined by the journal, which stipulate that “We recommend that all researchers who made substantial or important contributions to the design of a work, or the acquisition, analysis or interpretation of the data used in the paper, be included as authors.”, we would like to retain their authorship status. Furthermore, we ensured that all authors had read and approved the manuscript before submission, using Google Forms.

1. GUIDELINES ON GOOD PUBLICATION PRACTICE, Committee on Publication Ethics (COPE), https://publicationethics.org/files/u7141/1999pdf13.pdf

1. The last concern is about the significance of this study. Although the majority of strains showed increased lactate, some still showed decreased lactate levels in the brains. These results suggested that lactate or pH is an endophenotype for neuropsychiatric disorders, but it is hard to serve as a good diagnostic index as the change is not unidirectional in different disorders. In other words, the relationship between lactate level and neuropsychiatric disorders is not exclusive.

As pointed out by the reviewer, whether brain pH and lactate levels increase or decrease could vary among animal models. Such variation may represent subpopulations of patients or specific disease states. Considering both increases and decreases in changes in pH and lactate levels could be important to achieve that goal. Accordingly, we have revised the text as follows:

Added text (page 33, line 12)

"Detecting changes in brain pH and lactate levels, whether resulting in an increase or decrease due to their potential bidirectional alterations, using techniques such as MRS may help the diagnosis, subcategorization, and identification of specific disease states of these biologically heterogeneous and spectrum disorders, as has been shown for mitochondrial diseases (Lin et al., 2003)."

Added text (page 35, line 14)

"Therefore, while changes in brain pH and lactate levels could be transdiagnostic endophenotypes of neuropsychiatric disorders, they might occur depending on the subpopulation due to the distinct genetic and environmental causes or specific disease states in certain disorders."

**Reviewer #2 (Public Review):**
Hagihara et al. conducted a study investigating the correlation between decreased brain pH, increased brain lactate, and poor working memory. They found altered brain pH and lactate levels in animal models of neuropsychiatric and neurodegenerative disorders. Their study suggests that poor working memory performance may predict higher brain lactate levels.However, the study has some significant limitations. One major concern is that the authors examined whole-brain pH and lactate levels, which might not fully represent the complexity of disease states. Different brain regions and cell types may have distinct protein and metabolite profiles, leading to diverse disease outcomes. For instance, certain brain regions like the hippocampus and nucleus accumbens exhibit opposite protein/signaling pathways in neuropsychiatric disease models.

We want to thank the reviewer for the valuable suggestions. To address this issue, it is necessary to comprehensively identify brain regions and cell types responsible for pH and lactate changes in each strain/condition of animals, as these may differ among them. Subsequently, based on such findings, we can then proceed with functional investigations that specifically target the identified brain regions/cell types. However, conducting such investigations would require a significant amount of time to complete, approximately 2–3 years, and is beyond the scope of this study. Therefore, we would like to conduct such studies in the future. We have mentioned this limitation by revising the discussion section of this study as follows (page 43, line 5):

Original text

"Because we used whole brain samples to measure pH and lactate levels, we could not determine whether the observed changes in pH and/or lactate levels occurred ubiquitously throughout the brain or selectively in specific brain region(s) in each strain/condition of the models. Indeed, brain region-specific increases in lactate levels were observed in human patients with ASD in an MRS study (Goh et al., 2014). Furthermore, while increased lactate levels were observed in whole-brain measurements in mice with chronic social defeat stress (Figure S7) (Hagihara et al., 2021a), decreased lactate levels were found in the dorsomedial prefrontal cortex (Yao et al., 2023). The brain region-specific changes may occur even in animal models in which undetectable changes were observed in the present study. This could be due to the masking of such changes in the analysis when using whole-brain samples. Further studies are needed to address this issue by measuring microdissected brain samples and performing in vivo analyses using pH- or lactate-sensitive biosensor electrodes (Marunaka et al., 2014; Newman et al., 2011) and MRS (Davidovic et al., 2011)."

Revised text

"The major limitations of this study include the absence of analyses specific to brain regions or cell types and the lack of functional investigations. Because we used whole brain samples to measure pH and lactate levels, we could not determine whether the observed changes in pH and/or lactate levels occurred ubiquitously throughout the brain or selectively in specific brain region(s) in each strain/condition of the models. It is known that certain molecular expression profiles and signaling pathways display brain region-specific alterations, and in some cases, even exhibit opposing changes in neuropsychiatric disease models (Hosp et al., 2017; Floriou-Servou et al. 2018; Reim et al., 2017). Indeed, brain region-specific increases in lactate levels were observed in human patients with ASD in an MRS study (Goh et al., 2014). Furthermore, while increased lactate levels were observed in whole-brain measurements in mice with chronic social defeat stress (Figure S7) (Hagihara et al., 2021a), decreased lactate levels were found in the dorsomedial prefrontal cortex (Yao et al., 2023). Additionally, it has been reported that the basal intracellular pH differs between neurons and astrocytes (lower in astrocytes than in neurons), and their responsiveness to conditions simulating neural hyperexcitation and the metabolic acidosis in terms of intracellular pH also varies (Raimondo et al., 2016; Salameh et al., 2017). It would also be possible that the brain region/cell type-specific changes may occur even in animal models in which undetectable changes were observed in the present study. This could be due to the masking of such changes in the analysis when using whole-brain samples. Given the assumption that the brain regions and cell types responsible for pH and lactate changes vary across different strains/conditions, comprehensive studies are needed to thoroughly examine this issue for each animal model individually. This can be achieved through techniques such as evaluating microdissected brain samples, conducting in vivo analyses using pH- or lactate-sensitive biosensor electrodes (Marunaka et al., 2014; Newman et al., 2011), and MRS (Davidovic et al., 2011). Subsequently, based on such findings, it is also necessary to conduct functional analyses for each model animal by manipulating pH or lactate levels in specific brain regions/cell types and evaluating behavioral phenotypes relevant to neuropsychiatric disorders."

Moreover, the memory tests used in the study are specific to certain brain regions, but the authors did not measure lactate levels in those regions. Without making lactate measurements in brain-regions and cell types involved in these diseases, any conclusions regarding the role of lactate in CNS diseases is premature.

Regarding the point about “lactate measurements in brain-regions and cell types involved in these diseases,” please refer our responses provided above.

Additionally, evidence suggests that exogenous treatment with lactate has positive effects, such as antidepressant effects in multiple disease models (Carrard et al., 2018, Carrard et al., 2021, Karnib et al., 2019, Shaif et al., 2018). It also promotes learning, memory formation, neurogenesis, and synaptic plasticity (Suzuki et al., 2011, Yang et al., 2014, Weitian et al., 2015, Dong et al., 2017, El Hayek et al. 2019, Wang et al., 2019, Lu et al., 2019, Lev-Vachnish et a.l, 2019, Descalzi G et al., 2019, Herrera-López et al., 2020, Ikeda et al., 2021, Zhou et al., 2021,Roumes et al., 2021, Frame et al., 2023, Akter et al., 2023).

We thank the reviewer for pointing out many references regarding the effects of lactate that were not cited in our paper. We have since included these studies and discussed in more detail the effect of lactate at molecular, cellular, and behavioral levels (page 39, line 11).

Original text

"Moreover, increased lactate may have a positive or beneficial effect on memory function to compensate for its impairment, as lactate administration with an associated increase in brain lactate levels attenuates cognitive deficits in human patients (Bisri et al., 2016) and rodent models (Rice et al., 2002) of traumatic brain injury. In addition, lactate administration exerts antidepressant effects in a mouse model of depression (Carrard et al., 2016)."

Revised text

"Moreover, increased lactate may have a positive or beneficial effect on memory function to compensate for its impairment, as lactate administration with an associated increase in brain lactate levels attenuates cognitive deficits in human patients (Bisri et al., 2016) and rodent models (Rice et al., 2002) of traumatic brain injury. In addition, lactate administration exerts antidepressant effects in a mouse model of depression (Carrard et al., 2021, 2016; Karnib et al., 2019; Shaif et al., 2018). Lactate has also shown to promote learning and memory (Descalzi G et al., 2019; Dong et al., 2017; Hayek et al. 2019; Lu et al., 2019; Roumes et al., 2021; Suzuki et al., 2011), synaptic plasticity (Herrera-López et al., 2020; Yang et al., 2014; Zhou et al., 2021), adult hippocampal neurogenesis (Lev-Vachnish et al., 2019), and mitochondrial biogenesis and antioxidant defense (Akter et al., 2023), while its effects on adult hippocampal neurogenesis and learning and memory are controversial (Ikeda et al., 2021; Lev-Vachnish et al., 2019; Wang et al., 2019)."

In conclusion, the relevance of total brain pH and lactate levels as indicators of the observed correlations is controversial, and evidence points towards lactate having more positive rather than negative effects. It is important that the authors perform studies looking at brain-region-specific concentrations of lactate and that they modulate lactate levels (decrease) in animal models of disease to validate their conclusions. it is also important to consider the above-mentioned studies before concluding that "altered brain pH and lactate levels are rather involved in the underlying pathophysiology of some patients with neuropsychiatric disorders" and that "lactate can serve as a potential therapeutic target for neuropsychiatric disorders".

Regarding the points about positive effects of lactate, measurement of brain-region-specific lactate concentrations, and modulation of lactate levels, please refer to our responses provided earlier. The points raised by the reviewer are important and should be addressed in future studies.

**Reviewer #2 (Recommendations For The Authors):**
Measure lactate in specific brain regions. The whole brain measurements are not relevant to the disease states.

We thank the reviewer for pointing this out. We totally agree with the reviewer’s comment and recognize that the lack of investigations in specific brain regions is one of the major limitations of this study. To address this issue, it is necessary to comprehensively identify brain regions and cell types responsible for pH and lactate changes in each strain/condition of animals, as these may differ among them. Subsequently, based on such findings, we can then proceed with functional investigations that specifically target the identified brain regions/cell types. However, conducting such investigations would require a significant amount of time to complete, approximately 2–3 years, and is beyond the scope of this study. Therefore, we would like to conduct such studies in the future. We have mentioned this limitation by revising the discussion section of this study as follows (page 43, line 5):

Original text

"Because we used whole brain samples to measure pH and lactate levels, we could not determine whether the observed changes in pH and/or lactate levels occurred ubiquitously throughout the brain or selectively in specific brain region(s) in each strain/condition of the models. Indeed, brain region-specific increases in lactate levels were observed in human patients with ASD in an MRS study (Goh et al., 2014). Furthermore, while increased lactate levels were observed in whole-brain measurements in mice with chronic social defeat stress (Figure S7) (Hagihara et al., 2021a), decreased lactate levels were found in the dorsomedial prefrontal cortex (Yao et al., 2023). The brain region-specific changes may occur even in animal models in which undetectable changes were observed in the present study. This could be due to the masking of such changes in the analysis when using whole-brain samples. Further studies are needed to address this issue by measuring microdissected brain samples and performing in vivo analyses using pH- or lactate-sensitive biosensor electrodes (Marunaka et al., 2014; Newman et al., 2011) and MRS (Davidovic et al., 2011)."

Revised text:

"The major limitations of this study include the absence of analyses specific to brain regions or cell types and the lack of functional investigations. Because we used whole brain samples to measure pH and lactate levels, we could not determine whether the observed changes in pH and/or lactate levels occurred ubiquitously throughout the brain or selectively in specific brain region(s) in each strain/condition of the models. It is known that certain molecular expression profiles and signaling pathways display brain region-specific alterations, and in some cases, even exhibit opposing changes in neuropsychiatric disease models (Hosp et al., 2017; Floriou-Servou et al. 2018; Reim et al., 2017). Indeed, brain region-specific increases in lactate levels were observed in human patients with ASD in an MRS study (Goh et al., 2014). Furthermore, while increased lactate levels were observed in whole-brain measurements in mice with chronic social defeat stress (Figure S7) (Hagihara et al., 2021a), decreased lactate levels were found in the dorsomedial prefrontal cortex (Yao et al., 2023). Additionally, it has been reported that the basal intracellular pH differs between neurons and astrocytes (lower in astrocytes than in neurons), and their responsiveness to conditions simulating neural hyperexcitation and the metabolic acidosis in terms of intracellular pH also varies (Raimondo et al., 2016; Salameh et al., 2017). It would also be possible that the brain region/cell type-specific changes may occur even in animal models in which undetectable changes were observed in the present study. This could be due to the masking of such changes in the analysis when using whole-brain samples. Given the assumption that the brain regions and cell types responsible for pH and lactate changes vary across different strains/conditions, comprehensive studies are needed to thoroughly examine this issue for each animal model individually. This can be achieved through techniques such as evaluating microdissected brain samples, conducting in vivo analyses using pH- or lactate-sensitive biosensor electrodes (Marunaka et al., 2014; Newman et al., 2011), and MRS (Davidovic et al., 2011). Subsequently, based on such findings, it is also necessary to conduct functional analyses for each model animal by manipulating pH or lactate levels in specific brain regions/cell types and evaluating behavioral phenotypes relevant to neuropsychiatric disorders."

Discuss in detail the studies that show the neuroprotective effects of lactate and reconcile these with the authors' conclusions.

As suggested by the reviewer, we have discussed in more detail the positive effect of lactate at molecular, cellular, and behavioral levels as below (page 39, line 11):

Original text

"Moreover, increased lactate may have a positive or beneficial effect on memory function to compensate for its impairment, as lactate administration with an associated increase in brain lactate levels attenuates cognitive deficits in human patients (Bisri et al., 2016) and rodent models (Rice et al., 2002) of traumatic brain injury. In addition, lactate administration exerts antidepressant effects in a mouse model of depression (Carrard et al., 2016)."

Revised text

"Moreover, increased lactate may have a positive or beneficial effect on memory function to compensate for its impairment, as lactate administration with an associated increase in brain lactate levels attenuates cognitive deficits in human patients (Bisri et al., 2016) and rodent models (Rice et al., 2002) of traumatic brain injury. In addition, lactate administration exerts antidepressant effects in a mouse model of depression (Carrard et al., 2021, 2016; Karnib et al., 2019; Shaif et al., 2018). Lactate has also shown to promote learning and memory (Descalzi G et al., 2019; Dong et al., 2017; Hayek et al. 2019; Lu et al., 2019; Roumes et al., 2021; Suzuki et al., 2011), synaptic plasticity (Herrera-López et al., 2020; Yang et al., 2014; Zhou et al., 2021), adult hippocampal neurogenesis (Lev-Vachnish et al., 2019), and mitochondrial biogenesis and antioxidant defense (Akter et al., 2023), while its effects on adult hippocampal neurogenesis and learning and memory are controversial (Ikeda et al., 2021; Lev-Vachnish et al., 2019; Wang et al., 2019)."

Conduct experiments whereby you decrease/deplete/modulate lactate levels in animal models and show that there is amelioration of the symptoms.

Regarding this point, kindly refer to the responses we provided in the first comment from the reviewer. We have mentioned this limitation by revising the discussion section of this study as follows (page 43, line 5):

Original text

"Because we used whole brain samples to measure pH and lactate levels, we could not determine whether the observed changes in pH and/or lactate levels occurred ubiquitously throughout the brain or selectively in specific brain region(s) in each strain/condition of the models. Indeed, brain region-specific increases in lactate levels were observed in human patients with ASD in an MRS study (Goh et al., 2014). Furthermore, while increased lactate levels were observed in whole-brain measurements in mice with chronic social defeat stress (Figure S7) (Hagihara et al., 2021a), decreased lactate levels were found in the dorsomedial prefrontal cortex (Yao et al., 2023). The brain region-specific changes may occur even in animal models in which undetectable changes were observed in the present study. This could be due to the masking of such changes in the analysis when using whole-brain samples. Further studies are needed to address this issue by measuring microdissected brain samples and performing in vivo analyses using pH- or lactate-sensitive biosensor electrodes (Marunaka et al., 2014; Newman et al., 2011) and MRS (Davidovic et al., 2011)."

Revised text:

"The major limitations of this study include the absence of analyses specific to brain regions or cell types and the lack of functional investigations. Because we used whole brain samples to measure pH and lactate levels, we could not determine whether the observed changes in pH and/or lactate levels occurred ubiquitously throughout the brain or selectively in specific brain region(s) in each strain/condition of the models. It is known that certain molecular expression profiles and signaling pathways display brain region-specific alterations, and in some cases, even exhibit opposing changes in neuropsychiatric disease models (Hosp et al., 2017; Floriou-Servou et al. 2018; Reim et al., 2017). Indeed, brain region-specific increases in lactate levels were observed in human patients with ASD in an MRS study (Goh et al., 2014). Furthermore, while increased lactate levels were observed in whole-brain measurements in mice with chronic social defeat stress (Figure S7) (Hagihara et al., 2021a), decreased lactate levels were found in the dorsomedial prefrontal cortex (Yao et al., 2023). Additionally, it has been reported that the basal intracellular pH differs between neurons and astrocytes (lower in astrocytes than in neurons), and their responsiveness to conditions simulating neural hyperexcitation and the metabolic acidosis in terms of intracellular pH also varies (Raimondo et al., 2016; Salameh et al., 2017). It would also be possible that the brain region/cell type-specific changes may occur even in animal models in which undetectable changes were observed in the present study. This could be due to the masking of such changes in the analysis when using whole-brain samples. Given the assumption that the brain regions and cell types responsible for pH and lactate changes vary across different strains/conditions, comprehensive studies are needed to thoroughly examine this issue for each animal model individually. This can be achieved through techniques such as evaluating microdissected brain samples, conducting in vivo analyses using pH- or lactate-sensitive biosensor electrodes (Marunaka et al., 2014; Newman et al., 2011), and MRS (Davidovic et al., 2011). Subsequently, based on such findings, it is also necessary to conduct functional analyses for each model animal by manipulating pH or lactate levels in specific brain regions/cell types and evaluating behavioral phenotypes relevant to neuropsychiatric disorders."

Other corrections

Title page and Acknowledgements:

We have revised the affiliation information for the following co-authors: Drs. Anja Urbach8, Mohamed Darwish19, 20, Keizo Takao20, 22, Bong-Kiun Kaang53, 54, Michihiro Igarashi74, 75, Rie Ohashi87-89, and Nobuyuki Shiina87-89.

Page 56, line 12:

The term ‘The International Brain pH Consortium’ has been corrected to ‘The International Brain pH Project Consortium’.

Supplementary Table 1: Supplementary References:

1. Oota-Ishigaki A, Takao K, Yamada D, Sekiguchi M, Itoh M, Koshidata Y, et al. (2022): Prolonged contextual fear memory in AMPA receptor palmitoylation-deficient mice. Neuropsychopharmacology 47: 2150–2159.

We have updated the name of the mouse strain from “patDp” to “15q dup” throughout the manuscript.

We have made the following revisions to enhance readability.

Page 24, line 9: According to a simple correlation analysis, working memory measures (correct responses in the maze test) were significantly negatively correlated with brain lactate levels (r = -0.76, P = 1.93 × 10-5; Figure 1F).

Page 27, line 1:

Revised text

"We found that working memory measures (correct responses in the maze test) were the most frequently selected behavioral measures for constructing a successful prediction model (Figure 2E), which is consistent with the results of the exploratory study (Figure 1E)."

Figure 1 legend:

Revised text

"(F–H) Scatter plot showing correlations between actual brain lactate levels and measures of working memory (correct responses in the maze test) (F), the number of transitions in the light/dark transition test (G), and the percentage of immobility in the forced swim test (H)."

Figure 2 legend:

Revised text

"(F–H) Scatter plots showing correlations between actual brain lactate levels and working memory measures (correct responses in the maze test) (F), the acoustic startle response at 120 dB (G), and the time spent in dark room in the light/dark transition test (H)."

Page 30, line 2:

Original text

"The high to moderate-high pH/low to moderate-low lactate group included mouse models of ASD or developmental delay, such as Shank2 KO, Fmr1 KO, BTBR, Stxbp1 KO, Dyrk1 KO, Auts2 KO, and patDp mice (Table S1, Figure S7)."

Revised text

"The high pH/low lactate group and moderate-high pH/moderate-low lactate group included mouse models of ASD or developmental delay, such as Shank2 KO, Fmr1 KO, BTBR, Stxbp1 KO, Dyrk1 KO, Auts2 KO, and 15q dup mice (Table S1, Figure S7)."

Page 40, line 7:

Original text

"Moreover, increased lactate levels may also be involved in behavioral changes other than memory deficits such as anxiety."

Revised text

"Moreover, increased lactate levels may also be involved in behavioral changes other than memory deficits, such as anxiety."